# Identification, characterization, and structure-activity relationship of the ASIC3-selective peptide WRPRFa
Chun Chien [1,2], Nien-Du Yang [1,2], Victoria Jiang[1], Shanti M. Amagasu[1] & John M. Gilchrist [1] ✉

Acid-sensing ion channels (ASICs) are proton-gated cation channels that detect and signal increases in proton concentration. ASIC1a and ASIC3 play a role in pain sensation associated with extracellular acidification. There are few selective modulators of ASIC3, including the tetrapeptide RFamide RPRFa, which slows the acute desensitization of ASIC3. Here we describe the peptide WRPRFa as the most potent ASIC3 activator to date and a more effective pharmacological tool. WRPRFa enhances the pH sensitivity of ASIC3 and effectively removes acute desensitization. Additionally, we demonstrate that ASIC3 can undergo tachyphylaxis at very acidic pH, which is accelerated by WRPRFa. Our work characterizes a selective and effective in vitro tool to study the interaction of RFamides and ASICs, and by extension gating mechanisms of ASIC3.

Acid-sensing ion channels (ASICs) are trimeric, proton-gated cation channels[1]. They are part of the broader ENaC/Degenerin trimeric protein family and are also related to ligand-gated ion channels found broadly in marine invertebrates[2]. The ASIC family is composed of four genes, ASIC1-4, that give rise to six distinct isoforms: ASIC1a, 1b, 2a, 2b, 3, and 4. All isoforms respond to extracellular acidification except homomeric ASIC2b[3] and ASIC4[4]. ASIC1a and ASIC3 have a midpoint of activation around pH 6.5[1], while ASIC2 is closer to pH 4[5]. Their general structure is formed by 3 subunits, each with intracellular N- and C-termini and two transmembrane segments that contribute to the formation of a central ion-conducting pore. Between the two transmembrane segments is a large extracellular domain (ECD) whose structure has been described as a "closed fist" with the transmembrane domains connected to the ECD by a "wrist". The ECD contains a thumb, finger, knuckle, and palm domain, together "clutching" a β-ball domain[6]. Several structures have been solved for the chicken and human ASIC1a, as well as the related ENaC[7], FaNaC[8], and FaNaC1[9], but none for ASIC3.

Upon a decrease in extracellular pH, channels rapidly activate and open a sodium-selective pore, which gives rise to a depolarizing conductance. Activation is driven by protonation of charged Glu, Asp, and His residues distributed across the ECD, which causes a structural rearrangement of the ECD that opens the pore[10]. Shortly after activation channels undergo desensitization, where they remain closed until they recover at physiological pH[1]. Desensitization is driven by isomerization of the β11-12 linker in the lower palm domain[11]. The β11-12 linker acts as a molecular clutch that disengages the activated ECD from the pore, terminating the flow of sodium ions and leaving the channel unable to activate again until recovered. There are three recognized forms of

desensitization: (1) acute, (2) steady-state, closed-state, or resting-state, and (3) tachyphylaxis. Acute desensitization occurs after channel opening while steady-state desensitization (SSD) affects closed channels and occurs at sub-activating pHs[1]. Tachyphylaxis has been described only for ASIC1a and represents a progressive and nigh irrecoverable loss of current upon repeated stimulation[12,13].

ASICs are expressed in neurons of the central nervous system (CNS)[14], peripheral nervous system (PNS)[14,15], and the heart[16,17]. Correspondingly, studies have demonstrated their involvement in pain sensation[18,19], proprioception[20,21], fear and anxiety[22], synaptic plasticity[23], and cell death post-ischemia[24,25]. Both ASIC1a and ASIC3 have been shown to play a role in the proton sensitivity of pain-sensing neurons of the dorsal root ganglia (DRG)[26–28], although their respective contributions to pain signaling remain unclear. Dissecting the specific ASIC subunit composition of a neuron requires selective tool molecules, of which there are very few for ASIC3, including APETx2[29] and RPRFa[30]. APETx2 is an anemone toxin selective for ASIC3 over other ASIC isoforms but it can inhibit $Na_V1.8$[31] as well as ASIC3, complicating its use in nociceptive neurons where $Na_V1.8$ is widely expressed. RPRFa is an RFamide peptide from the venom of *Conus textile*, a predatory marine snail, and selectively slows the desensitization rate of ASIC3 to prolong the open state. Other RFamides exhibit lower margins of selectivity, like FRRFa and FMRFa[32], or can exert opposite effects on different isoforms, like As2a[33].

RFamides are a class of signaling peptides characterized by an Arg-Phe motif with an amidated C-terminus[34]. RFamides are found across almost all levels of animal life, from FMRFa-gated sodium channels[35] in mollusks to mammalian neuropeptides[36,37], some of which have been linked to nociception and analgesia[38,39]. Multiple RFamides with

[1]Latigo Biotherapeutics, Inc, Thousand Oaks, CA, USA. [2]These authors contributed equally: Chun Chien, Nien-Du Yang. ✉e-mail: jgilchrist@latigobio.com

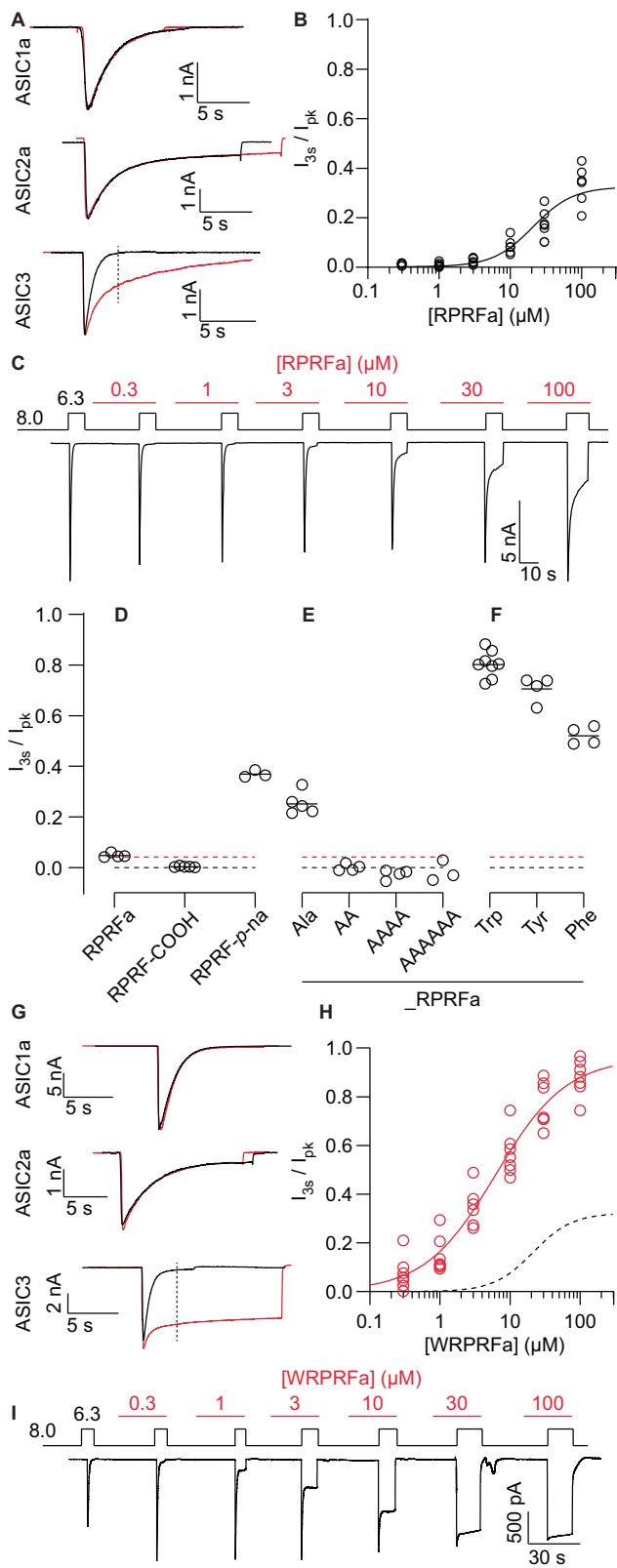

**Fig. 1 | N-terminal extension increases the activity of RPRFa. A** Example traces of the effect of 30 μM RPRFa (red) on ASIC1a, ASIC2a, and ASIC3 in response to an acid stimulus (pH 6.3 for ASIC1a and ASIC3, pH 4.0 for ASIC2a), tested by manual patch clamp. Dashed line on ASIC3 trace shows the time 3 s after peak ($I_{3s}$), which was used to calculate $I_{3s}/I_{pk}$. **B** Concentration-response curve for RPRFa on ASIC3 tested on MPC. The y-axis, $I_{3s}/I_{pk}$, displays the ratio of the current remaining after 3 s to the peak current as a measure of peptide activity. Data collected from six cells. **C** Example protocol (top) and trace (bottom) from a cell plotted in (**B**). Steps indicate change in pH, red shows application of RPRFa. Effect of RPRFa and C-terminal modified peptides (**D**) N-terminal extension with alanines (**E**) and aromatic amino acids (**F**) tested on the QPatch II. Black dashed lines indicates 0 (no change) and red dashed lines show the average response of RPRFa. Solid black lines depict geometric mean. **G** Traces showing the effect of 30 μM WRPRFa (red) on ASIC1a, ASIC2a, and ASIC3 during acid stimulation (pH 6.3 for ASIC1a and ASIC3, pH 4.0 for ASIC2a), tested by MPC. **H** Concentration-response curve of WRPRFa on ASIC3 tested on MPC. Dotted line depicts the RPRFa concentration-response curve shown in (**B**). Data collected from seven cells. **I** Example protocol (top) and trace (bottom) from a cell plotted in (**H**). Steps indicate change in pH, red shows application of WRPRFa.

create challenges for studying their interaction with ASICs and the transient nature of the interaction further limits their use. An improvement to the pharmacological properties of RPRFa would enable a clearer understanding of the interaction between RFamides and ASICs and enhance their utility as tools to study ASIC biology.

Here we describe the RPRFa derivative WRPRFa, which shows increased potency and efficacy compared to its parent peptide. Using an automated patch-clamp screen of synthetic peptides, we characterized the SAR for each amino acid of the peptide. Mutagenesis of WRPRFa and the ASIC3 lower palm domain confirmed a specific intermolecular contact between the peptide and channel. WRPRFa stabilizes the activated state of ASIC3, increasing its sensitivity to protons and dramatically slowing the rate of desensitization. Furthermore, we demonstrate ASIC3 tachyphylaxis and show WRPRFa accelerates ASIC3 tachyphylaxis both from activated and desensitized states.

## Results

### N-terminal extension of RPRFa increases its efficacy

We generated a panel of custom peptides to test on ASIC3 to characterize the SAR of RPRFa. Peptides were designed to introduce a variety of changes for each amino acid to understand their contribution to RPRFa activity. To confirm the activity of the synthesized peptides, we tested RPRFa against ASIC3 stably expressed in human embryonic kidney (HEK) cells by manual patch clamp (MPC). All electrophysiology experiments were performed by MPC except for those indicated to have been done on QPatch II. RPRFa slowed the desensitization for ASIC3 but not ASIC2a, with a small effect on ASIC1a (Fig. 1A, Fig. S1). We chose pH 6.3 as our standard stimulus pH because it is near the midpoint of pH-dependence of activation ($pH_{50}$) for ASIC3, which would allow us to detect both increases and decreases in current. ASIC3 desensitization at pH 6.3 is rapid and complete, with negligible current remaining 3 s after activation. We used the current remaining 3 s after the peak as a measure of the current originating from RPRFa-bound channels, and normalization to the peak current gave an approximation of the fraction of channels bound to peptide ($I_{3s}/I_{pk}$) (Fig. 1A, bottom)[30,45]. RPRFa had an $EC_{50}$ with a lower limit of 21.3 μM (95% CI: 16.6–27.1 μM) against ASIC3 (Fig. 1B and C), higher than previously published (4.23 μM)[30]. The discrepancy in $EC_{50}$ could arise from species or methodological differences: our study measured the fraction of sustained current at 3 s post-peak using human ASIC3 while the prior study used rat ASIC3 and measured sustained current at 5 s.

Next, we tested the effect of our peptides on ASIC3 stably expressed in HEK or Chinese hamster ovary (CHO) cells using the QPatch II automated electrophysiology platform. Both cell lines displayed comparable gating properties; we observed only slight differences in their sensitivity to pH or rate of desensitization (Fig. S2A and B). Peptides tested on ASIC3 expressed in CHO cells are presented in Tables S1 and S2. HEK cells are known to

divergent N-terminal sequences have shown activity on ASIC3 but little work has been done to understand their structure-activity relationship (SAR)[40–42]. Structural studies reveal FaNaCs bind their ligand FMRFa in the distal finger domain but docking and mutagenesis studies indicate FRRFa and RPRFa bind the lower palm domain in ASIC1a[43,44] and ASIC3[45], respectively. The small effects of natural RFamides, like RPRFa,

**Table 1 | Activity of WRPRFa derivatives on ASIC3 sustained current**

| Peptide | Geometric mean ($I_{3s}/I_{pk}$) | Geometric S.D. factor | N | *p*-value |
|---|---|---|---|---|
| WRPRFa | 0.806 | 1.07 | 8 | |
| WRPRF-COOH | 0.249 | 1.12 | 6 | <0.0001 |
| Site 1 | | | | |
| WRPR{Cha}a | 0.756 | 1.03 | 5 | 0.994 |
| WRPR{Phe(F5)}a | 0.742 | 1.04 | 5 | 0.8964 |
| WRPR{HomoPhe}a | 0.727 | 1.06 | 4 | 0.706 |
| WRPRWa | 0.693 | 1.08 | 6 | 0.0484 |
| WRPR{Cpg}a | 0.660 | 1.02 | 4 | 0.0092 |
| WRPRIa | 0.619 | 1.02 | 4 | 0.0001 |
| WRPRYa | 0.554 | 1.11 | 4 | <0.0001 |
| WRPR{Tic}a | 0.421 | 1.07 | 6 | <0.0001 |
| WRPRAa | 0.201 | 1.29 | 5 | <0.0001 |
| WRPRHa | 0.149 | 1.17 | 4 | <0.0001 |
| Site 2 | | | | |
| WRP{HomoArg}Fa | 0.744 | 1.03 | 6 | 0.868 |
| WRP{Arg(Me)}Fa | 0.656 | 1.05 | 4 | 0.0066 |
| WRP{SDMA}Fa | 0.510 | 1.06 | 5 | <0.0001 |
| WRP{ADMA}Fa | 0.428 | 1.14 | 4 | <0.0001 |
| WRPKFa | 0.167 | 1.11 | 4 | <0.0001 |
| WRPIFa | 0.0958 | 1.29 | 4 | <0.0001 |
| WRPNFa | 0.085 | 1.20 | 5 | <0.0001 |
| WRPQFa | 0.0361 | 1.09 | 4 | <0.0001 |
| WRPAFa | 0.0154 | 1.91 | 5 | <0.0001 |
| WRP{Cit}Fa | 0.0228 | 1.77 | 6 | <0.0001 |
| WRPDFa | 0.00475 | 1.68 | 4 | <0.0001 |
| WRPEFa | 0.00363 | 3.47 | 4 | <0.0001 |
| Site 3 | | | | |
| WRSRFa | 0.633 | 1.16 | 4 | 0.0011 |
| WRARFa | 0.623 | 1.08 | 6 | <0.0001 |
| WR{Tic}RFa | 0.595 | 1.39 | 4 | 0.0002 |
| WRNRFa | 0.395 | 1.08 | 4 | <0.0001 |
| WRQRFa | 0.355 | 1.52 | 7 | <0.0001 |
| WRGRFa | 0.237 | 1.52 | 8 | <0.0001 |
| Site 4 | | | | |
| W{HomoArg}PRFa | 0.722 | 1.04 | 4 | 0.582 |
| WKPRFa | 0.583 | 1.03 | 5 | <0.0001 |
| WIPRFa | 0.400 | 1.29 | 7 | <0.0001 |
| WMPRFa | 0.248 | 1.20 | 5 | <0.0001 |
| W{Cit}PRFa | 0.180 | 1.12 | 4 | <0.0001 |
| WAPRFa | 0.0844 | 1.16 | 4 | <0.0001 |
| WEPRFa | 0.0432 | 1.14 | 4 | <0.0001 |
| WDPRFa | 0.0166 | 1.12 | 4 | <0.0001 |
| Site 5 | | | | |
| YRPRFa | 0.705 | 1.08 | 4 | 0.285 |
| FRPRFa | 0.521 | 1.07 | 4 | <0.0001 |
| {HomoPhe}RPRFa | 0.451 | 1.10 | 4 | <0.0001 |
| {Phe(4-NH2)}RPRFa | 0.436 | 1.18 | 4 | <0.0001 |
| IRPRFa | 0.362 | 1.24 | 4 | <0.0001 |
| {Cpg}RPRFa | 0.350 | 1.09 | 4 | <0.0001 |

**Table 1 (continued) | Activity of WRPRFa derivatives on ASIC3 sustained current**

| Peptide | Geometric mean ($I_{3s}/I_{pk}$) | Geometric S.D. factor | N | *p*-value |
|---|---|---|---|---|
| {Cpa}RPRFa | 0.346 | 1.05 | 4 | <0.0001 |
| {Tic}RPRFa | 0.335 | 1.16 | 4 | <0.0001 |
| {Cha}RPRFa | 0.276 | 1.42 | 4 | <0.0001 |
| ARPRFa | 0.251 | 1.18 | 5 | <0.0001 |
| {Phe(F5)}RPRFa | 0.213 | 1.21 | 6 | <0.0001 |
| RRPRFa | 0.174 | 1.21 | 4 | <0.0001 |
| VRPRFa | 0.157 | 1.38 | 5 | <0.0001 |
| HRPRFa | 0.166 | 1.59 | 4 | <0.0001 |
| ERPRFa | 0.00440* | 0.0104* | 5 | <0.0001 |

*arithmetic mean ± S.D.
Peptides were tested at 30 μM against ASIC3 on the QPatch II to understand their structure-activity relationship. Geometric mean and geometric S.D. are reported for $I_{3s}/I_{pk}$ after peptide treatment. P-value represents multiplicity adjusted p-value from Dunnett's multiple comparisons test, versus WRPRFa. Peptides are ordered by descending activity.

express endogenous ASIC1, which could form heteromers with the stably expressed ASIC3 and alter pharmacology[46,47]. We performed a qRT-PCR experiment to measure transcript copy number and determined that the level of ASIC3 transcript was 307-fold greater than ASIC1, indicating heteromers would constitute ~1% of the total channel population (Fig S2C–E). Therefore, the level of endogenous ASIC1 in HEK cells is insufficient to affect the interpretation of results.

For the screen, we used $I_{3s}/I_{pk}$ as the measure of peptide activity on ASIC3. Cells were incubated with 3 or 30 μM peptide for at least 2 min at pH 8.0, followed by a 3 s application of pH 6.3 without peptide (Fig. S3A and B). Most of the 31 peptides tested showed partial or total loss of activity (Tables S1 and S2). The modest activity of 30 μM RPRFa ($I_{3s}/I_{pk}$: 0.0476 ± 0.00838) made it difficult to accurately determine changes in activity. However, we confirmed the requirement for C-terminal amidation, without which the peptide was inactive (RPRF-COOH $I_{3s}/I_{pk}$: 0.00339 ± 0.00259) (Fig. 1D) and furthermore observed that replacement of the C-terminal amide with a para-nitroanilide (p-na) increased the effect of RPRFa (RPRF-p-na $I_{3s}/I_{pk}$: 0.387 ± 0.028).

Inspired by RPRF-p-na, we tested if RPRFa could be changed to increase its activity. Given that C-terminal amidation is crucial, we chose to extend the peptide from the N-terminus. The addition of a single Ala increased its effect (ARPRFa $I_{3s}/I_{pk}$: 0.251 ×/÷ 1.18), but further addition of alanines removed all activity (Fig. 1E, Table 1, Table S1). Changing the methyl side chain of Ala into the aromatic ring of Trp, Tyr, or Phe caused a striking increase of activity (Fig. 1F, Table 1). The effect was most pronounced with WRPRFa (structure shown in Fig. S3C), with 80% of the channels ($I_{3s}/I_{pk}$: 0.806 ×/÷ 1.07) affected by 30 μM WRPRFa.

Throughout the rest of this study, we continued to use the $I_{3s}/I_{pk}$ ratio to estimate the fraction of bound channels when evaluating WRPRFa activity. WRPRFa maintains high selectivity for ASIC3 over ASIC1a and ASIC2a, neither of which showed significant change in sustained current after WRPRFa treatment (Fig. 1G, Fig. S4). WRPRFa is both more potent and more effective than RPRFa, with almost 90% of channels affected at 100 μM and an $EC_{50}$ of 6.22 μM (95% CI: 4.27–10.8 μM), which is 3-fold more potent than the lower limit for RPRFa (Fig. 1H and I).

## SAR identifies key residues for peptide activity

Given the small signal window for RPRFa, we reapproached our SAR investigation using WRPRFa as a background because it allowed us to discern smaller changes to activity than was possible with RPRFa. The results of the screen are recorded in Table 1. Structures of non-canonical amino acids are displayed in Fig. S5.

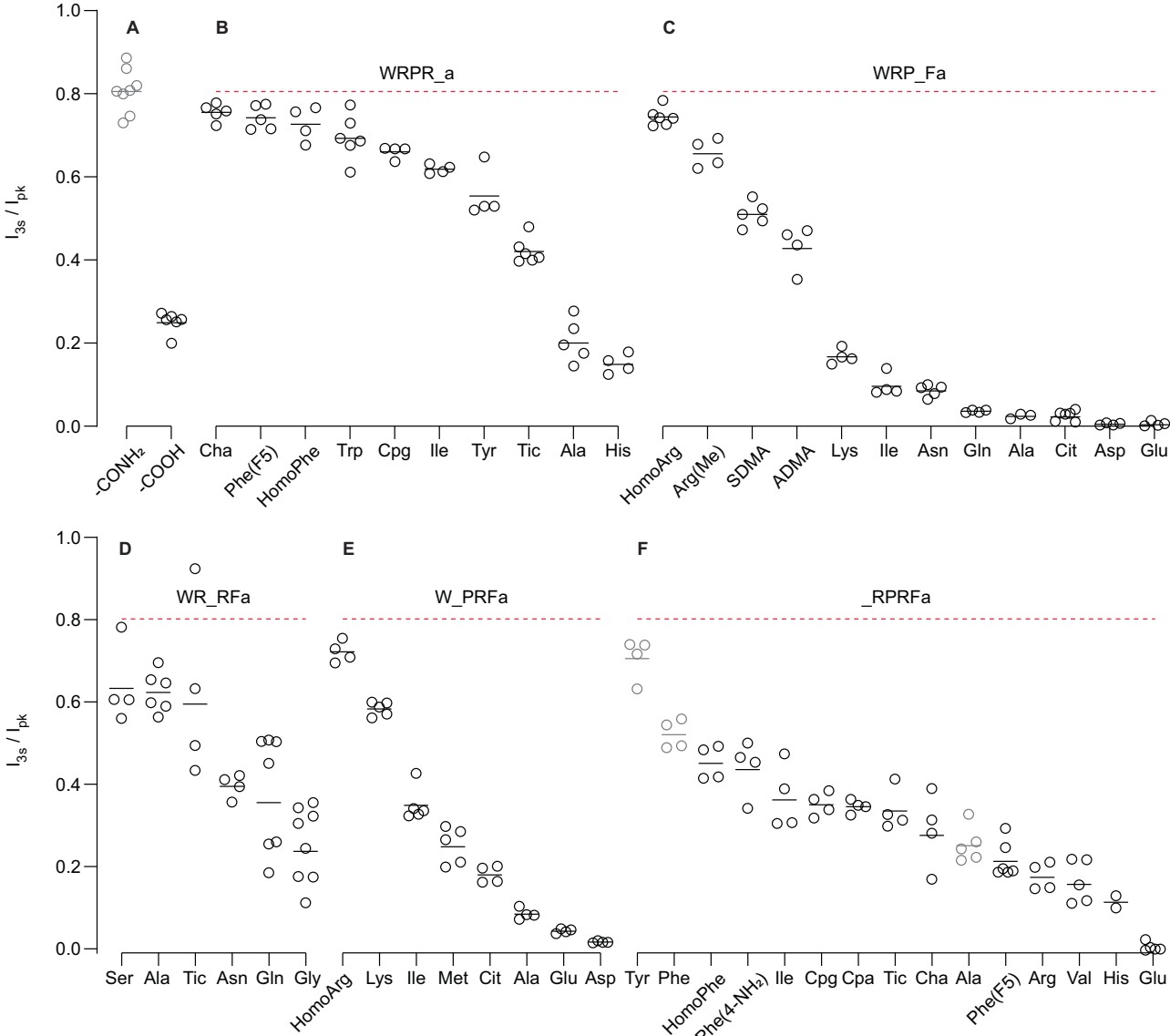

**Fig. 2 | Structure-activity relationship of WRPRFa.** Graphs plotting the $I_{3s}/I_{pk}$ for changes to the C-terminus (**A**) and sites 1–5 (**B–F**). Dashed red line indicates the geometric mean activity of 30 μM WRPRFa. Gray points for ARPRFa, WRPRFa, FRPRFa, and YRPRFa are replotted from Fig. 1E and F. Solid black lines depict geometric mean.

Although the C-terminal amidation is required for maximal effect, it was not necessary for partial activity of the WRPRF-COOH peptide ($I_{3s}/I_{pk}$: 0.249 ×/÷ 1.12) (Fig. 2A). As the C-terminus was fixed, we used the C-terminal Phe as our index for numbering; for WRPRFa, the C-terminal Phe is site 1 while the N-terminal Trp is site 5. Variants are plotted by descending activity in Fig. 2.

For site 1, Phe could be replaced by almost every other ringed side chain amino acid while maintaining activity comparable to WRPRFa (Fig. 2B). Modifications to the phenyl ring like fluorination (Phe[F5]) or linker extension (HomoPhe) had similar effects. A ring is not strictly necessary; the bulky, hydrophobic Ile was comparable to the ringed amino acids. The small methyl side chain of Ala decreased activity substantially, perhaps due to the loss of hydrophobic bulk. While most peptides had diminished activity at 3 μM (Table S3), dearomatization of the phenyl ring in WRPR{Cha}a exhibited greater activity than the WRPRFa parent peptide. We tested the potency of this peptide and WRPRFa on the QPatch II and found WRPR{Cha}a was more potent (WRPR{Cha}a EC50: 2.33 μM, 95% CI: 2.05–2.67 μM; WRPRFa EC50: 6.31 μM, 95% CI: 5.93–6.72 μM; >99.9% EC50 are different) (Fig. S6). His

was the least effective of those tested, perhaps owing to the partial positive charge of its imidazole side chain.

Replacement of the canonical Arg at site 2 resulted in a total or nearly total loss of activity (Fig. 2C). Extension of the Arg side chain did not impact activity (WRP{HomoArg}Fa), but increasing methylation of the guanidine group further diminished the effect (WRP{Arg(Me)}Fa, WRP{SDMA}Fa, WRP{ADMA}Fa). Lys possesses a positively charged side chain like Arg, but only retains a small amount of activity.

Site 3 was relatively accommodating of differing side chains. Proline, due to its constrained nature, is often essential for establishing secondary structures in proteins, but replacement of Pro with Ser or Ala had only a small effect on activity (Fig. 2D). Larger side chains like Asn, Gln, or the isoquinoline of Tic showed decreased activity but not to the same degree as changes of the site 2 Arg. On the other hand, the very small Gly with a proton side chain was the least effective of those tested. Together, these results suggest a preference for small side chains connected to a chiral carbon.

Changes to the site 4 Arg revealed a greater tolerance than site 2 (Fig. 2E). The charged amino acid Lys decreased activity but not as much as the corresponding change at site 2 (WRPKFa). Furthermore, the activity of

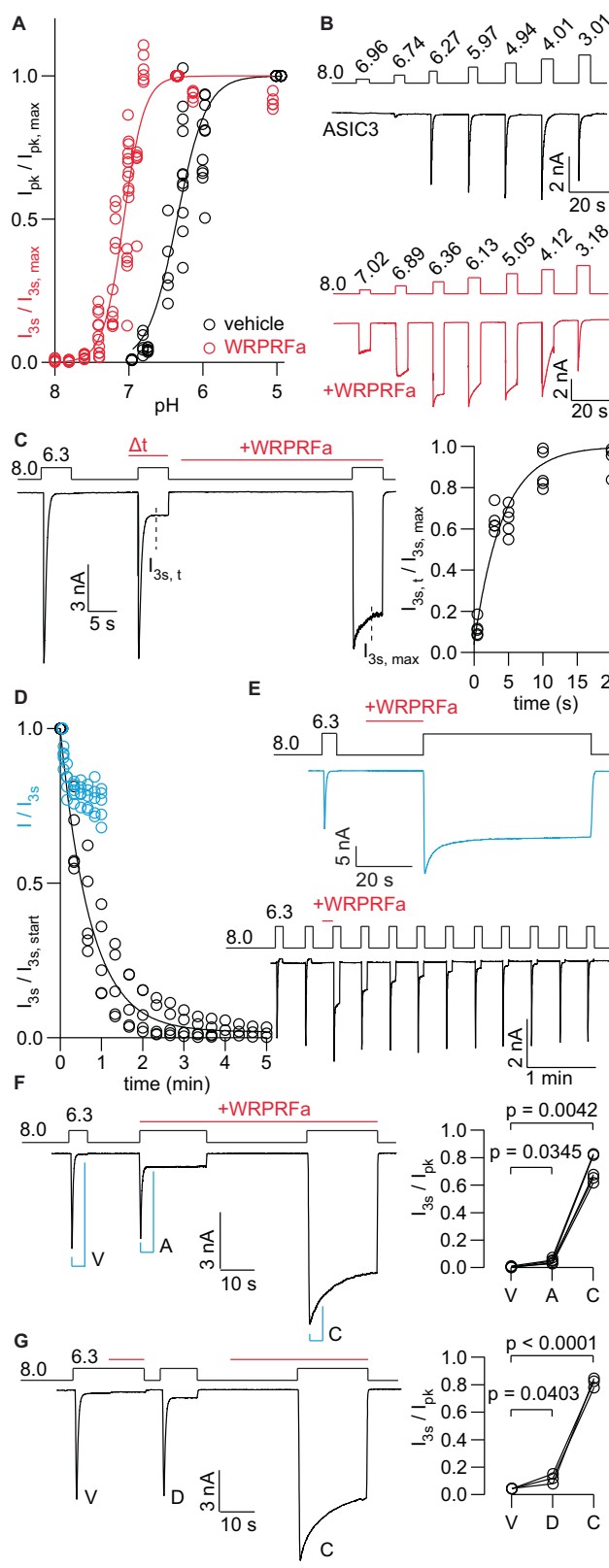

**Fig. 3 | WRPRFa increases the pH-sensitivity of ASIC3 and stabilizes the activated state. A** pH-dependence of activation for ASIC3 without (black) or with (red) 30 μM WRPRFa. Current from vehicle-treated channels is measured as $I_{pk}$ and from peptide-treated channels as $I_{3s}$. Data collected from 11 cells for vehicle-treated and 18 cells for WRPRFa-treated. **B** Representative traces of data depicted in (**A**). Protocol is shown above each trace; steps indicate change in pH. Red trace indicates entire protocol was performed in the presence of 30 μM WRPRFa. **C** Time-dependence of WRPRFa binding to closed-state channel. Left, a trace showing application of 30 μM WRPRFa applied for increasing durations of time. A final step with 30 s application time to achieve maximum effect was used for normalization. Right, graph showing the rate of WRPRFa interaction with the closed state of ASIC3, fit with a monoexponential decay function. Data collected from 25 cells. **D** Time-dependence of WRPRFa wash off from ASIC3 using pH 6.3 (blue) or pH 8.0 (black). Data were fit with a monoexponential decay function. Data collected from five cells for pH 8.0 wash off and 5 cells for pH 6.3 wash off. **E** Traces of data from (**D**). Top shows the current decay over 1 min of pH 6.3 stimulus. Bottom shows wash off using pH 8.0. Every 20 s a 5 second pH 6.3 stimulus was given to measure the fraction of sustained current remaining as indicator of bound WRPRFa. Above each trace, steps in the protocol indicate change in applied pH and red bars show when 30 μM WRPRFa was applied to cells. **F** Left shows protocol (above) and trace (below) depicting binding of 30 μM WRPRFa to the activated state of ASIC3. V (vehicle), A (activated), and C (closed) represent specific channel states and blue bars where measurements were taken. Right, graph showing the sustained current ($I_{3s}/I_{pk}$) before treatment (V), with co-application of WRPRFa (A), and to the closed state (C). Data collected from five cells. **G** Desensitized state binding of 30 μM WRPRFa, following same outline as (**F**). D indicates the desensitized state. Data collected from three cells. Significance in (**F**) and (**G**) were calculated with two-tailed ratio paired t-test.

(Fig. 2F). Many of the derivatives were modifications to a phenyl ring and so can be compared directly with Phe (FRPRFa). Extending the linker of Phe (HomoPhe), adding an amine (Phe[4-NH₂]), or fluorinating (Phe[F5]) it all decreased activity. Dearomatization of the Phe phenyl ring in Cha decreased activity, and cyclopentane (Cpg) and cyclopropane (Cpa) were comparably affected. As in site 1, His was ineffective; Arg and Glu were also much less effective. Together, these results indicate a preference for flatter, bulky hydrophobic groups to engage in π-stacking interactions.

Lastly, increasing the WRPRFa peptide further in length with another N-terminal ringed side chain amino acid caused activity to decrease, suggesting the larger peptides may be exceeding the bounds of the ligand pocket or otherwise diminishing their activity (Fig. S7).

## WRPRFa stabilizes the activated state of ASIC3

Given the increased activity of WRPRFa relative to RPRFa, we tested if its effects on ASIC3 gating would also be increased. RPRFa has previously been reported to induce a slight alkalinizing shift (0.09 units) in the pH-dependence of activation for ASIC3[30]. We performed a similar experiment: HEK cells stably expressing ASIC3 were held at pH 8.0 and stimulated for 3 s with increasingly acidic solutions. For untreated channels, the peak current ($I_{pk}$) was measured and normalized to pH 5.0. To assess the properties of RPRFa- or WRPRFa-bound channels, we performed the same experiment in the presence of 30 μM RPRFa or 30 μM WRPRFa and reported the $I_{3s}$ normalized to pH 6.3 for WRPRFa and pH 5.0 for RPRFa. We observed a shift of 0.73 units in the alkaline direction after WRPRFa treatment (vehicle pH₅₀: 6.34, 95% CI: 6.30–6.39; WRPRFa pH₅₀: 7.07, 95% CI: 7.04–7.09; >99.9% pH₅₀ are different) (Fig. 3A and B). Consistent with the prior report, we found RPRFa shifted the pH₅₀ by 0.10 units (RPRFa pH₅₀: 6.44, 95% CI: 6.39–6.50; vs vehicle, 63.5% pH₅₀ are different) (Fig. S8A).

As WRPRFa caused an alkaline shift in the pH-sensitivity of ASIC3 activation, treatment with the peptide should increase the current magnitude in response to a fixed acid stimulus. The pH-dependence of activation described above indicates pH 6.3 activated about 55.7% of unbound channels and 99.7% of WRPRFa-bound channels, which should result in a 2-fold increase in current after WRPRFa binding. We examined the fold-change in peak current after WRPRFa treatment during the experiment

Ile and Met was a little less than that of Lys, suggesting the length of the side chain may be as important as the cationic charge. However, the anionic side chains of Glu and Asp lost almost all activity, suggesting a fundamental incompatibility with the ligand pocket.

Exploration of site 5 indicated Trp was the most effective amino acid of those evaluated, although Tyr was roughly comparable in efficacy

depicted in Fig. 1F. The measured fold-change was 2.26 ×/÷ 1.19 (Fig. S8B), consistent with our prediction.

We then determined the rate at which WRPRFa interacts with ASIC3. To measure the peptide on-rate to the resting state, we applied 30 μM WRPRFa at pH 8.0 for increasing durations, immediately followed by a test pulse at pH 6.3 to measure the sustained current (Fig. 3C). After this test pulse, we incubated cells with 30 μM WRPRFa for 30 s to elicit the maximum effect and used this value for normalization. Interaction of WRPRFa with resting state channels was completed within 20 s, with a time constant of 3.35 s (95% CI: 2.39–4.70 s). WRPRFa was applied with a focal perfusion system that changed the bath solution around the cell in ~100 ms, much faster than the rate at which WRPRFa binds to the channel and therefore should not influence the apparent on-rate to the resting state channels.

Next, we assessed the rate at which WRPRFa leaves the channel from the activated and closed states. We preincubated cells with WRPRFa to load channels with peptide, then used two different protocols to remove the peptide. For dissociation from the activated state, we preincubated cells at pH 8.0 with 30 μM WRPRFa, then continuously applied pH 6.3 without WRPRFa to activate the channels while washing off bound peptide. HEK cells poorly tolerate long periods of acid application and sodium influx, so acid application was kept to 1 min. A decrease of about 22% occurred within the first 10 s with a time constant of 11.3 s (95% CI: 8.10–16.2 s), after which the current remained nearly constant (Fig. 3D and E). The first decrease of current likely arose from desensitization of unbound channels, and possibly also from an initial dissociation of peptide from activated channels, which subsequently desensitized.

For dissociation from resting state channels, we continuously applied pH 8.0 without peptide to remove WRPRFa. Every 20 s the cells were given a 5 s application of pH 6.3 to quantify the remaining sustained current. A 5 s activation was used to minimize wash-off from the activated state, which was deemed negligible based on the observed off-rate under the pH 6.3 condition. The sustained current was about 95% gone after 3 min, with a time constant of 43.5 s (95% CI: 39.0–48.4 s) (Fig. 3D and E).

We next asked if WRPRFa displays any state-preference. ASIC3 can occupy a resting state, an activated state, or a desensitized state, any of which may favor WRPRFa binding. To assess activity on the activated state of ASIC3, we co-applied 30 μM WRPRFa with an activating pH 6.3 stimulus for 20 s, followed by a recovery interval at pH 8.0 in the presence of WRPRFa (Fig. 3F). Untreated cells showed an $I_{3s}/I_{pk}$ of 0.00339 ×/÷ 3.96, and 0.0448 ×/÷ 1.44 after co-application of WRPRFa with the pH 6.3 stimulus, indicating a small but significant increase in the sustained current (p = 0.0345, two-tailed ratio paired t-test). During the recovery interval, the cells were exposed to 30 μM WRPRFa for 20 s, after which the $I_{3s}/I_{pk}$ increased to 0.714 ×/÷ 1.14.

We performed a similar experiment to assess binding to the desensitized state (Fig. 3G). After channel activation and desensitization, we applied 30 μM WRPRFa to the still-desensitized channels for 10 s to allow for binding. Next the cells were held at pH 8.0 without peptide to recover from desensitization and stimulated without peptide at pH 6.3 to measure the extent of binding. The $I_{3s}/I_{pk}$ ratio after binding to desensitized channels was 0.113 ×/÷ 1.38 (vs. vehicle $I_{3s}/I_{pk}$: 0.0439 ×/÷ 1.03; p = 0.0403, two-tailed ratio paired t-test), much less than that observed for resting-state channels in the same experiment (0.817 ×/÷ 1.04). Collectively, these results suggest WRPRFa preferentially binds the closed or resting state but once bound dissociates much more slowly from the activated state than the resting state. Given the reduced off-rate from the activated state and increased sensitivity to protons, our results suggest WRPRFa stabilizes the activated state of ASIC3.

**A glutamate pair in the lower palm domain forms part of the WRPRFa interaction site**

Next, we sought to determine if the interaction site for WRPRFa differs from that of previously reported RFamides like FRRFa[43,44] and RPRFa[45]. FRRFa and RPRFa have been virtually docked to ASIC1a and ASIC3, respectively, and mutagenesis studies have confirmed their interaction with the lower

palm domain of ASICs. Here we set out to determine if WRPRFa interacts in the same region. Additionally, we asked if we could leverage the results of our SAR study to validate sidechain interactions proposed by prior docking studies.

From our peptide mutagenesis screen, we saw the largest changes to WRPRFa activity came from charge-neutralizing mutations of the site 2 and site 4 Arg residues. Given the importance of these cations, we looked for anionic amino acids in the lower palm domain as potential interaction partners. On the β sheets forming the palm domain in ASIC3 we found several Glu residues (Fig. 4A). As there is no structure of ASIC3, we examined the orientation of the corresponding residues in the chicken ASIC1a, which are conserved between channels (Fig. 4A). Around the β11-12 linker, which is crucial for desensitization, we saw four glutamates: two above the β11-12 linker (E378 and E416) and two below (E78 and E421) (Fig. 4B). The lower two glutamates (E78/E421) have been previously described to form a pair, with their side chains in close approximation[6,48].

We individually mutated the four glutamates to glutamine to remove their charge but maintain the side chain structure (Fig. 4C). We tested each mutant by incubating HEK cells stably expressing ASIC3 or CHO cells transfected with mutant ASIC3 with 30 μM WRPRFa before a pH 6.3 acid stimulus. E78Q showed a response to WRPRFa that resembled the WT channel, although the currents under vehicle conditions were much smaller before treatment (Fig. 4D), which suggested the mutation altered channel gating but retained its capacity for interaction with WRPRFa. The activity of WRPRFa on E378Q and E416Q was unchanged compared to the WT channel, indicating these two glutamates may not play a significant role in the ASIC3/WRPRFa interaction. The E421Q mutation attenuated the effect of WRPRFa on channel desensitization (Fig. 4E). We also mutated each of the four glutamates to aspartate but were unable to detect measurable currents. This could be due to poor expression or problems of surface trafficking, or because the proton sensitivity of the mutant channels was shifted such that the pH 6.3 stimulus we used was insufficient to activate the channels.

The residues E79 (ASIC3: E78) and E416 (ASIC3: E421) in ASIC1a have previously been shown to participate in pH gating, where E to Q mutation of either residue decreases the pH sensitivity of the channel[48]. We asked if the same was true for ASIC3 and could explain the increase in E78Q peak current after WRPRFa treatment. We measured the pH-dependence of activation for E78Q and E421Q by activating channels with increasingly acidic stimuli. The E78Q mutant channel is less sensitive to acid than the WT channel, with a $pH_{50}$ of 6.10 (95% CI: 6.03–6.16; WT $pH_{50}$: 6.34; >99.9% $pH_{50}$ are different) (Fig. 4F), while E421Q is similar to WT ASIC3 (E421Q $pH_{50}$: 6.33, 95% CI: 6.27–6.38; WT $pH_{50}$: 6.34; 26.5% $pH_{50}$ are different) (Fig. 4G). We treated E78Q with 30 μM WRPRFa and observed the peptide shifted the E78Q $pH_{50}$ of activation like the WT channel (E78Q/WRPRFa $pH_{50}$: 7.03, 95% CI: 7.00–7.07; WT/WRPRFa $pH_{50}$: 7.07; 46.5% $pH_{50}$ are different) (Fig. 4F). These results indicate the increase of E78Q peak current after WRPRFa treatment is due to a heightened sensitivity to pH induced by the peptide.

Given that changes to proton sensitivity might affect the apparent effect of WRPRFa, we measured the $EC_{50}$ of WRPRFa on each of the four mutants by treating cells with escalating concentrations of WRPRFa before activation with pH 6.3; the results are presented in Table 2. WRPRFa potency on the E78Q mutant increased 30-fold compared to WT (Fig. 5A). Potency on E378Q (Fig. 5B) and E416Q (Fig. 5C) was slightly decreased or increased, respectively, while the E421Q mutant markedly decreased WRPRFa activity (Fig. 5D).

We then asked which of the two WRPRFa arginines interacts with the E78/E421 pair. A prior study of RPRFa and ASIC3 proposed an interaction of the site 2 Arg with E78/E421 and although their mutagenesis experiments confirmed the importance of both glutamates, specific interaction with the site 2 Arg could not be demonstrated by single mutagenesis[45]. To this end, we used a double mutant cycle approach to identify coupling between site 2 or site 4 of WRPRFa with the E78/E421 pair. In this case, the double mutant cycle experiment compared the change in potency caused by single

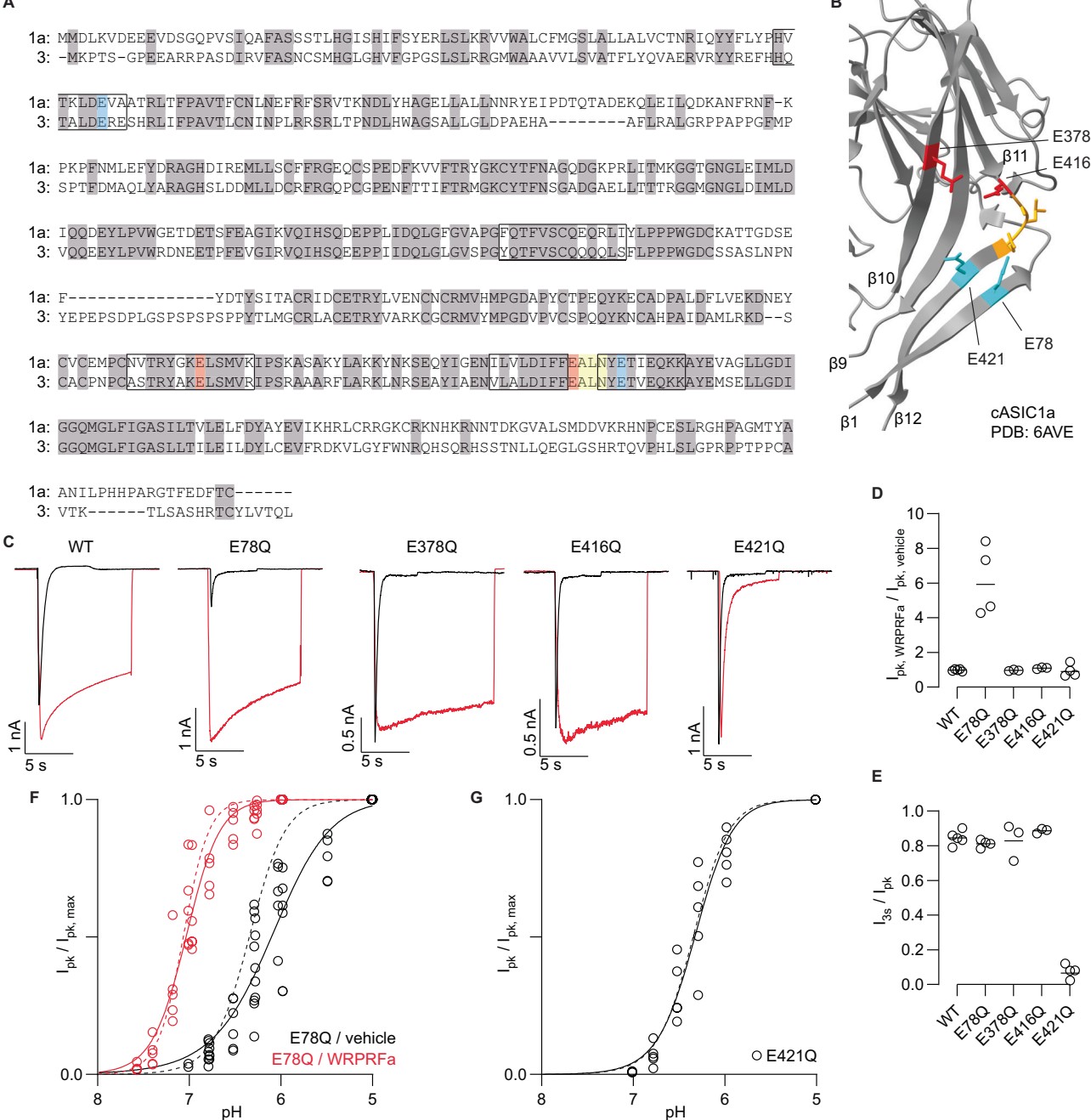

**Fig. 4 | Mutations in the lower palm domain affect ASIC3 pH-sensitivity and affect WRPRFa activity. A** Alignment of chicken ASIC1a (1a) and ASIC3 (3) showing conserved residues highlighted in gray, the β11-12 linker in yellow, and mutated palm domain glutamates in red and cyan. β-sheets composing the palm domain are indicated by black boxes. **B** Structure of cASIC1a (PDB: 6AVE) in resting state. The β11-12 linker is colored yellow, the upper glutamates in red, and the lower glutamate pair cyan. **C** Representative traces of WT ASIC3, E78Q, E378Q, E416Q, and E421Q before (black) and after (red) 30 μM WRPRFa application, in response to a pH 6.3 stimulus. **D** Graph depicting the fold-increase in peak current ($I_{pk}/I_{pk}$) after 30 μM WRPRFa treatment. Data collected from four cells for E78Q, three cells for E378Q, three cells for E416Q, four cells for E421Q, and five cells for WT. Solid black lines depict geometric mean. **E** Graph depicting the $I_{3s}/I_{pk}$ for each channel shown in (**C**). Data collected from the same cells as in (**D**). Solid black lines depict geometric mean. **F** pH-dependence of activation for E78Q before (black) and after (red) treatment with 30 μM WRPRFa. Data collected from 11 cells for vehicle-treated and 10 for WRPRFa-treated. Dashed lines indicate the pH-dependence of activation curves for WT ASIC3 before (black) and after (red) treatment with WRPRFa, replotted from Fig. 3A. **G** pH-dependence of activation for the E421Q mutant. Data collected from five cells. Dashed line indicates the pH-dependence of activation curve for WT ASIC3, replotted from Fig. 3A.

mutations in either WRPRFa or ASIC3 against the change in potency caused by simultaneous mutations in WRPRFa and ASIC3. A coupling coefficient (Ω) can be calculated by dividing the product of the WT/WRPRFa potency and double mutant potency by the product of the two single mutant potencies. An uncoupled pair has a coefficient near 1, while Ω > 2 suggests functional coupling[49,50].

We chose E78Q because, unlike E421Q, its potency on ASIC3 could be fully resolved. Given the proximity of the side chains in the cASIC1a structure it is probable these glutamates function as a pair[6,48]. Thus, mutation of either glutamate should impact peptide interactions with both side chains. For site 4 on WRPRFa we used WIPRFa and site 2 WRP{ADMA}Fa, as both changes decreased the activity by about half (Table 1). WIPRFa was at least

**Table 2 | Potency of RFamides on WT and mutant ASIC3**

| Channel | Peptide | EC$_{50}$ (95% CI) | E$_{max}$ (95% CI) | Hill slope (95% CI) | n / concentration |
|---|---|---|---|---|---|
| WT | WRPRFa | 6.22 (4.27–10.8) | 0.959 (0.862–1.12) | 0.860 (0.667–1.09) | 6–7 |
| WT[a] | WRPRFa | 6.31 (5.93–6.72) | 0.736 (0.721–0.753) | 1.44 (1.34–1.55) | 6 |
| WT[a] | WIPRFa | 24.0 (21.5–26.7) | 0.570[b] | 1.79 (1.53–2.12) | 3–4 |
| WT | WRP{ADMA}Fa | 21.3 (15.9–33.3) | 0.852 (0.752–undefined) | 1.22 (0.936–1.58) | 6 |
| E78Q | WRPRFa | 0.206 (0.176–0.242) | 0.826 (0.795–0.860) | 1.40 (1.11–1.75) | 6 |
| E78Q | WIPRFa | 1.66 (0.909–6.82) | 0.729 (0.631–0.990) | 0.729 (0.410–1.17) | 6 |
| E78Q | WRP{ADMA}Fa | 4.93 (2.57–23.1) | 0.524 (0.439–0.761) | 0.800 (0.481–1.25) | 5 |
| E378Q | WRPRFa | 9.59 (6.65–15.0) | 0.902 (0.800–undefined) | 0.993 (0.770–1.28) | 5–6 |
| E378Q | WIPRFa | 20.8 (9.95–25.6) | 1.00 (0.787–undefined) | 0.856 (0.707–1.29) | 5 |
| E416Q | WRPRFa | 2.47 (1.71–3.89) | 0.846 (0.762–0.963) | 1.30 (0.840–2.10) | 6 |
| E416Q | WIPRFa | 10.3 (6.60–23.2) | 0.846 (0.729–undefined) | 0.982 (0.697–1.39) | 5 |
| E421Q | WRPRFa | 22.5 (12.3–38.3) | 0.283[b] | 1.494 (0.772–undefined) | 6 |

[a]experiment performed on QPatch II; [b]E$_{max}$ constrained

Peptides were tested at escalating concentrations against ASIC3 by manual patch clamp or on the QPatch II determine their potency against WT or mutant ASIC3. Activity was measured with I$_{3s}$/I$_{pk}$ and parameters reported with 95% confidence intervals.

4-fold less potent than WRPRFa on WT ASIC3 and 8-fold less potent than WRPRFa on E78Q (Table 2, Fig. 5E). WRP{ADMA}Fa was 3-fold less potent on WT and 24-fold less potent on E78Q (Table 2, Fig. 5F). For WIPRFa, the upper limit of $\Omega$ was 2.09 (Fig. 5G), though the true value is likely lower as the constrained fit for WIPRFa potency on WT ASIC3 overestimates its potency. For WRP{ADMA}Fa, $\Omega = 6.99$ (Fig. 5H), indicating functional coupling between WRPRFa site 2 and E78.

Having confirmed the site 2 Arg interaction with the E78/E421 pair, we next asked if the site 4 Arg interaction with the rat ASIC3 E418 (hASIC3: E416)[45] could be determined by the same method. E378 and E416 have not been shown to function as a pair, so an arginine sidechain may interact only with one glutamate. As we already established the interaction of the site 2 Arg with E78/E421, we only tested for coupling between the site 4 Arg with E378 and E416 using WIPRFa. The potency shift of WIPRFa on both mutant channels was similar (Table 2), and for site 4 Arg and E416 we calculated a $\Omega \le 1.08$, indicative of absent coupling between these sites. However, the interaction between the site 4 Arg and E416 may be too weak to measure with our approach.

**WRPRFa differentially affects ASIC1a/ASIC3 heteromers**

Given the E78/E421 pair is not located at a subunit-subunit interface, it may be possible that most of or the entire WRPRFa binding site could be provided by a single subunit. This would result in three potential binding sites on a single ASIC3 trimer. Structures of FMRFa bound to FaNaC and FaNaC1 show three peptides per homomeric channel, while modeling of the heteromeric HyNaC suggested only a single RFa-I[51] binds per channel, at a specific inter-subunit interface. However, these sites are distinct from the lower palm domain that constitutes the RFamide binding site in ASICs. The concentration-response curve for WRPRFa against ASIC3 has a Hill slope of 0.860, which was not indicative of multiple sites involved in cooperative binding. We sought to understand if a single ASIC3 subunit would be sufficient for WRPRFa activity and if the peptide would still be effective in a heteromeric channel containing non-ASIC3 subunits.

Prior studies of RFamide interaction with ASIC heteromers have used co-injection into oocytes, which does not allow isolation of a single heteromeric species[30,52–54]. To overcome this limitation, we generated heteromeric concatemer channels of ASIC3 and ASIC1a subunits, using the latter as insensitive replacements for binding-competent ASIC3 subunits. This is an approach that has been used successfully in the past to generate diverse heteromeric ASICs[5,55,56]. We generated plasmid constructs encoding three ASIC subunits in tandem, with inter-subunit N- and C-termini connected by a six amino acid linker: AGSVDS between the first and second subunits and AGSGSS between the second and third subunits. By sandwiching an

ASIC1a subunit between N- and C-terminal ASIC3 subunits, we made the ASIC313 channel. Likewise, inserting an ASIC3 subunit between two ASIC1a subunits formed the ASIC131 channel (Fig. 6A).

ASIC313 and ASIC131, when transfected into WT CHO cells, expressed robustly and displayed gating properties similar to ASIC3. Both heteromeric channels showed desensitization after activation by pH 6.3 that was faster than ASIC1a and comparable to ASIC3 (Fig. 6B and C). The pH-dependence of activation was 0.07 units different between the two heteromers (ASIC313 pH$_{50}$: 6.50, 95% CI: 6.47–6.54; ASIC131 pH$_{50}$: 6.43, 95% CI: 6.39–6.46; 98.3% pH$_{50}$ are different) and distinct from ASIC3 (WT ASIC3 pH$_{50}$: 6.34; ASIC313 vs ASIC3, 96.4% pH$_{50}$ are different; ASIC131 vs ASIC3, >99.9% pH$_{50}$ are different) (Fig. 6D and E).

Next, we assessed the impact of altered subunit composition on peptide activity. Preincubation of cells with 30 μM WRPRFa induced a 0.26 unit alkaline shift in the pH-dependence of activation for ASIC131 (ASIC131/WRPRFa pH$_{50}$: 6.69, 95% CI: 6.65–6.72; vs ASIC3, >99.9% pH$_{50}$ are different) and 0.38 units for ASIC313 (ASIC313/WRPRFa pH$_{50}$: 6.88, 95% CI: 6.85–6.91; vs ASIC3, >99.9% pH$_{50}$ are different) (Fig. 6D and E), suggesting that peptide modification of only a single subunit is sufficient to alter pH gating for the entire channel. For untreated cells, the peak current (I$_{pk}$) was recorded, and for WRPRFa-treated cells, the current 3 s after the peak (I$_{3s}$) was used. The shift in pH-sensitivity for ASIC313 was less than WT ASIC3 (0.38 units vs 0.73 units; ASIC313 vs WT ASIC3, >99.9% pH$_{50}$ shift is different), as was the shift for ASIC131 (0.26 units vs 0.73 units; ASIC131 vs WT ASIC3, >99.9% pH$_{50}$ shift is different). ASIC131 was shifted even less than ASIC313 (0.26 units vs 0.38 units; ASIC131 vs ASIC313, >99.9% pH$_{50}$ shift is different), consistent with the number of ASIC3 subunits in the channel.

Furthermore, the EC$_{50}$ for WRPRFa also changed with the number of ASIC3 subunits in the channel. WRPRFa was less potent on ASIC313 (lower limit EC$_{50}$: 12.6 uM, 95% CI: 10.2–15.6 μM; vs ASIC3, >99.9% EC$_{50}$ are different) than ASIC3 and even less so on ASIC131 (lower limit EC$_{50}$: 32.4 μM, 95% CI: 28.5–36.5 μM; vs ASIC3, >99.9% EC$_{50}$ are different) (Fig. 6F and G). With each loss of an ASIC3 subunit, the EC$_{50}$ decreased approximately 2-fold. Taken together, these results suggest a single ASIC3 subunit provides a sufficient WRPRFa binding site but do not necessarily imply multiple peptides can bind to a single channel.

**WRPRFa affects the rate and pH-dependence of ASIC3 desensitization**

ASIC3 exhibits both acute desensitization and SSD. Acute desensitization is pH-independent and terminates channel activation, while SSD occurs at sub-activating pH levels without apparent channel activation. Additionally,

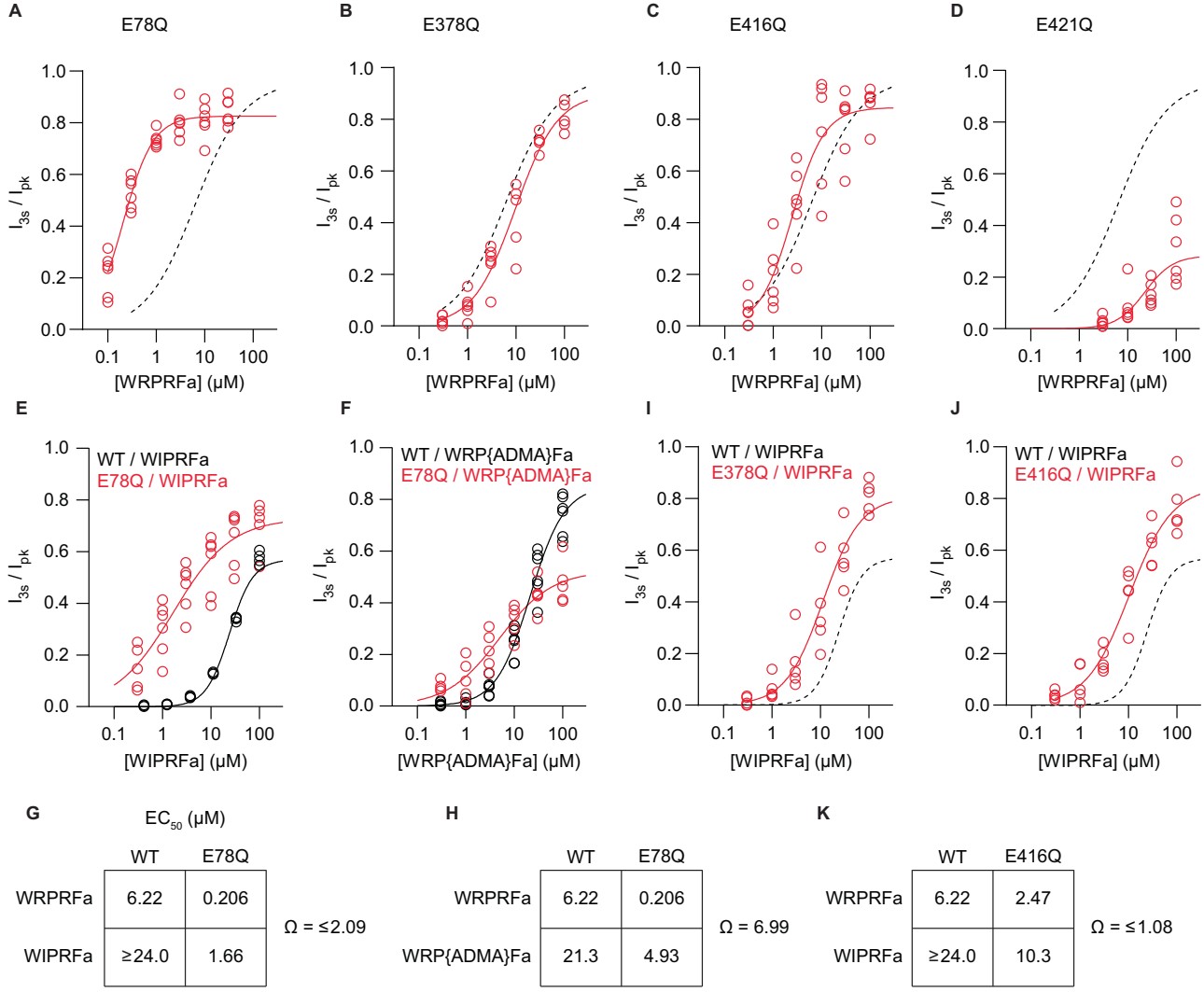

**Fig. 5 | The E78/E421 glutamate pair interacts with the site 2 Arg of WRPRFa.** Concentration-response curve of WRPRFa on E78Q (**A**), E378Q (**B**), E416Q (**C**), and E421Q (**D**) depicted in red, with the concentration-response curves for WRPRFa shown as a black dashed line, replotted from Fig. 1H. Data collected from six cells for all mutants. **E** Concentration-response curves of WIPRFa on WT ASIC3 (black) and E78Q (red). Data collected from four cells for WT and six cells for E78Q. **F** Concentration-response curves of WRP{ADMA}Fa on WT ASIC3 (black) and E78Q (red). Data collected from six cells for WT and five cells for E78Q. **G, H** display a quadrant box containing $EC_{50}$ values for each combination of peptide and channel used to calculate the coupling coefficient $\Omega$. Concentration-response curves for WIPRFa on E378Q (**I**) and E416Q (**J**) on WT ASIC3 (black) and mutant (red). Data collected from five cells for E378Q and E416Q. **K** A quadrant box containing $EC_{50}$ values for each combination of peptide and channel used to calculate the coupling coefficient $\Omega$ for the E416Q and WIPRFa mutations.

SSD shows pH-dependence, where desensitization becomes more complete as the stimulus grows more acidic[1].

Here, we measured the effect of WRPRFa on the acute desensitization rate of ASIC3 stably expressed in HEK cells. Channels were activated with acidic stimuli ranging from pH 7.4 to 4.0 and the rate of desensitization was determined by fitting a monoexponential decay function. Cells pre-treated with 30 μM WRPRFa were held at the test pH for 30 s to enable sufficient decay for a reliable fit. In the absence of peptide, ASIC3 acute desensitization is rapid, with a time constant under 1 s and consistently fast from pH 6.5 to 4.0 (Fig. 7A and B). Treatment with WRPRFa, however, dramatically slowed the desensitization rate most prominently at more alkaline pH and progressively accelerated as the activating stimulus became more acidic. At pH 4.0 the rate of acute desensitization for WRPRFa-treated ASIC3 is within 2-fold of the unbound channels (vehicle τ: 552 ms ×/÷ 1.49; WRPRFa τ: 893 ms ×/÷ 1.25; $p = 0.0067$, two-tailed unpaired t-test) but at pH 6.3 WRPRFa-bound channels are closer to 100 times slower (vehicle τ: 260 ms ×/÷ 1.36, WRPRFa τ: 23,400 ms ×/÷ 1.20; $p < 0.0001$, two-tailed unpaired t-test).

Prior work characterizing the interaction of RPRFa with ASIC3 established that RPRFa must first dissociate from the channel before desensitization could occur[45]. The apparent pH-sensitivity of acute desensitization for WRPRFa-treated channels could therefore reflect the titration of a charged sidechain by protons that affects the strength of peptide binding. We performed an experiment to ask if WRPRFa would function the same way as RPRFa[45]; that is, the peptide must first dissociate from the channel before desensitization can occur. HEK cells expressing ASIC3 were pre-treated with 30 μM WRPRFa to load channels with peptide, then activated by a pH 5.0 solution without peptide, which induced a relatively rapid desensitization. If WRPRFa must first unbind from channels before desensitization, then a continuous flow of pH 5.0 solution would wash away the peptide that dissociated from channels prior to desensitization. Once most of the current was gone, we recovered the channels for 2 s at pH 8.0, which allowed unbound and desensitized channels to partially recover (Fig. S9A). After the brief recovery interval, we resumed the pH 5.0 stimulus to re-activate unbound channels that recovered during the 2 s interval, as well as channels still bound to the peptide.

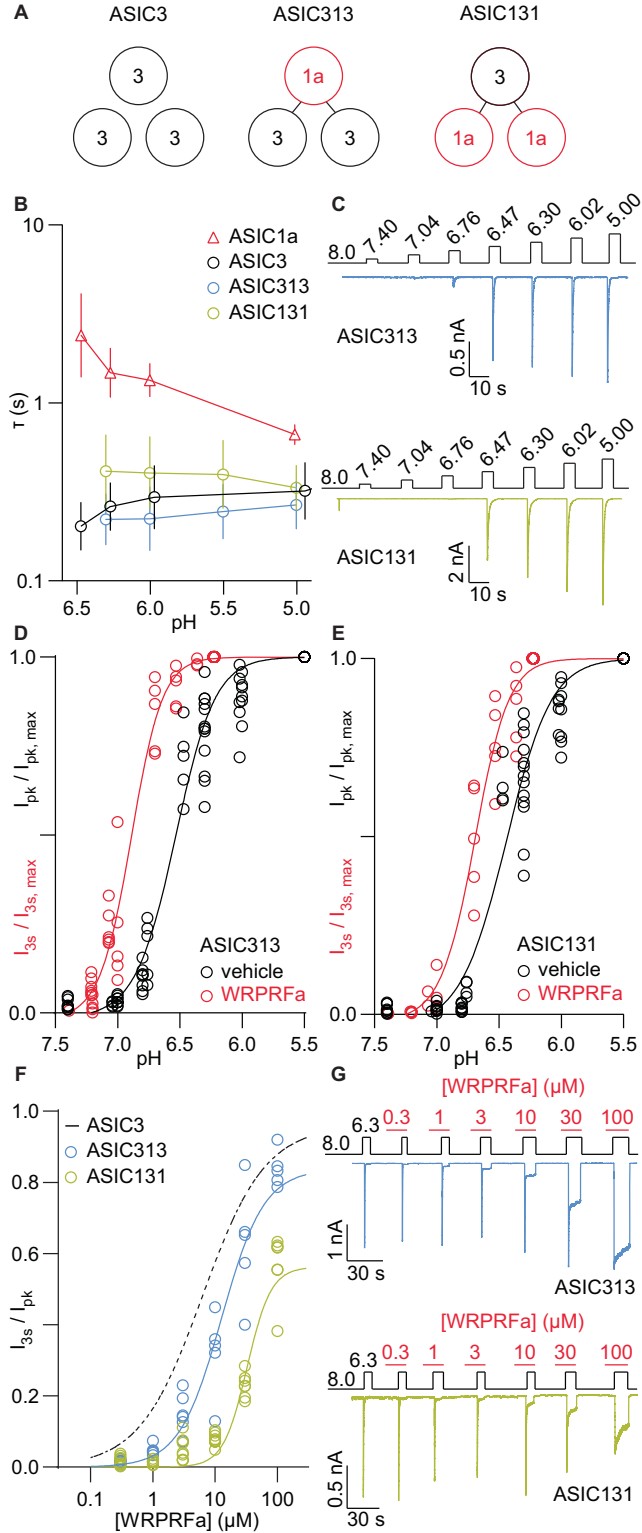

**Fig. 6 | WRPRFa activity titrates with ASIC3 subunits. A** Diagram showing subunit composition of ASIC3 and the concatemeric heterotrimers ASIC313 and ASIC131. Black circles indicate ASIC3 subunits and red ASIC1a subunits. **B** pH-dependence of acute desensitization for ASIC1a (red), ASIC3 (black), ASIC313 (blue), and ASIC131 (mustard). Data collected from four cells for ASIC1a, 11 cells for ASIC3, eight cells for ASIC313, and nine cells for ASIC131. Symbols represent geometric mean and error bars geometric standard deviation. **C** Protocol (above) and traces (below) for cells plotted in (**B**), (**D**), and (**E**). Top is ASIC313 and bottom ASIC131. Steps in the protocol represent change in pH. **D** pH-dependence of activation of ASIC313, before (black) and after (red) application of 30 μM WRPRFa. Current from vehicle-treated channels is reported as $I_{pk}$ and from peptide-treated channels as $I_{3s}$. Data collected from 13 cells for vehicle-treated and 11 cells for WRPRFa-treated. **E** pH-dependence of activation of ASIC131, before (black) and after (red) application of 30 μM WRPRFa. Current from vehicle-treated channels is reported as $I_{pk}$ and from peptide-treated channels as $I_{3s}$. Data collected from 13 cells for vehicle-treated and 11 cells for WRPRFa-treated. **F** Concentration-response curves of WRPRFa against ASIC313 (blue) and ASIC131 (mustard). Dashed black line is for ASIC3, replotted from Fig. 1H. Data collected from six cells for ASIC313 and eight cells for ASIC131. **G** Protocol (above) and traces (below) for ASIC313 (top) and ASIC131 (bottom) from cells plotted in (**F**). Red bars indicate application of WRPRFa.

and our study, peak ii showed a decreased sustained current fraction compared to peak i, reflecting an increased proportion of the fast-desensitizing component originating from unbound channels.

We considered these data in the context of a previously published gating scheme (Fig. 7E). Desensitization in the presence of peptide exhibits a fast and slow component; the fast component ($\tau_f$) arises from the acute desensitization of unbound channels and the slow component ($\tau_s$) was suggested to be the unbinding of RPRFa from ASIC3, shown in blue in Fig. 7E. The prior study identified no occupancy of the peptide-bound, desensitized state ($D_B$)[45]. On the other hand, we did not observe recovery of unbound channels, suggesting the slow component of desensitization of WRPRFa-treated channels may be a bona fide desensitization of bound channels (Fig. 7E, red). Additionally, the peak current was not fully recovered after a 20 s recovery period at pH 8.0 in the presence of WRPRFa, indicative of very slow recovery from the $D_B$ state (Fig. S9D).

Lastly, we tested the effect of 30 μM WRPRFa on SSD of ASIC3. First, we sought to determine if, like activation, WRPRFa also changed the pH-dependence of steady-state desensitization for ASIC3. We preconditioned cells at pH 8.0–7.0 for 20 s to elicit SSD, followed immediately by a test pulse to pH 6.3 (Fig. 7F and G) to measure the fraction of channels that were not desensitized. The current evoked by each pH 6.3 test pulse was normalized to the pH 6.3 pulse following pH 8.0 conditioning and plotted against the conditioning pH to determine the pH-dependence of SSD. For untreated cells, the peak current ($I_{pk}$) was recorded, and for WRPRFa-treated cells, the current 3 s after the peak ($I_{3s}$) was used. Untreated channels were completely desensitized at pH 7.0, while the WRPRFa-treated channels reached a plateau at pH 7.0 with about 30% of channels desensitized. Furthermore, untreated cells exhibited a pH-dependence of desensitization with $pH_{50}$ of 7.49 (95% CI: 7.47–7.51), while WRPRFa-treated cells were acid-shifted to a $pH_{50}$ of 7.38 (95% CI: 7.27–7.44; vs untreated, 99.1% $pH_{50}$ are different) (Fig. 7D). The 0.11 unit shift is larger than that reported for RPRFa (0.03 unit shift)[30].

## ASIC3 exhibits tachyphylaxis at low pH

During our experiments, we observed that ASIC3 currents would become smaller or disappear entirely after successive activations at pH 4.0. This effect recalled the irrecoverable loss of current seen during ASIC1a tachyphylaxis. Upon repeated acid stimulation, ASIC1a current gradually decreases and cannot be recovered. The rate at which it does is dependent on the activating pH as well as time spent in the activated/open state. Tachyphylaxis has not been reported for WT ASIC3, although one study indicated a partial effect at pH 5.6 in the N414K mutant of rat ASIC3, which increases open time by slowing desensitization[57].

If WRPRFa dissociated from channels before desensitization, then the peptide would have been washed away during the peak i stimulus period. Thus, fewer channels would be peptide-bound in peak ii and the fraction of sustained current would be lower for peak ii than for peak i. Unexpectedly, we saw no difference in the $I_{3s}/I_{pk}$ between peaks i and ii, indicating that desensitization of WRPRFa-bound channels did not result in an increased number of unbound channels (Fig. 7D). This is a different outcome than was seen for RPRFa (Fig. S9B and C)[45]. For RPRFa, both in the previous study

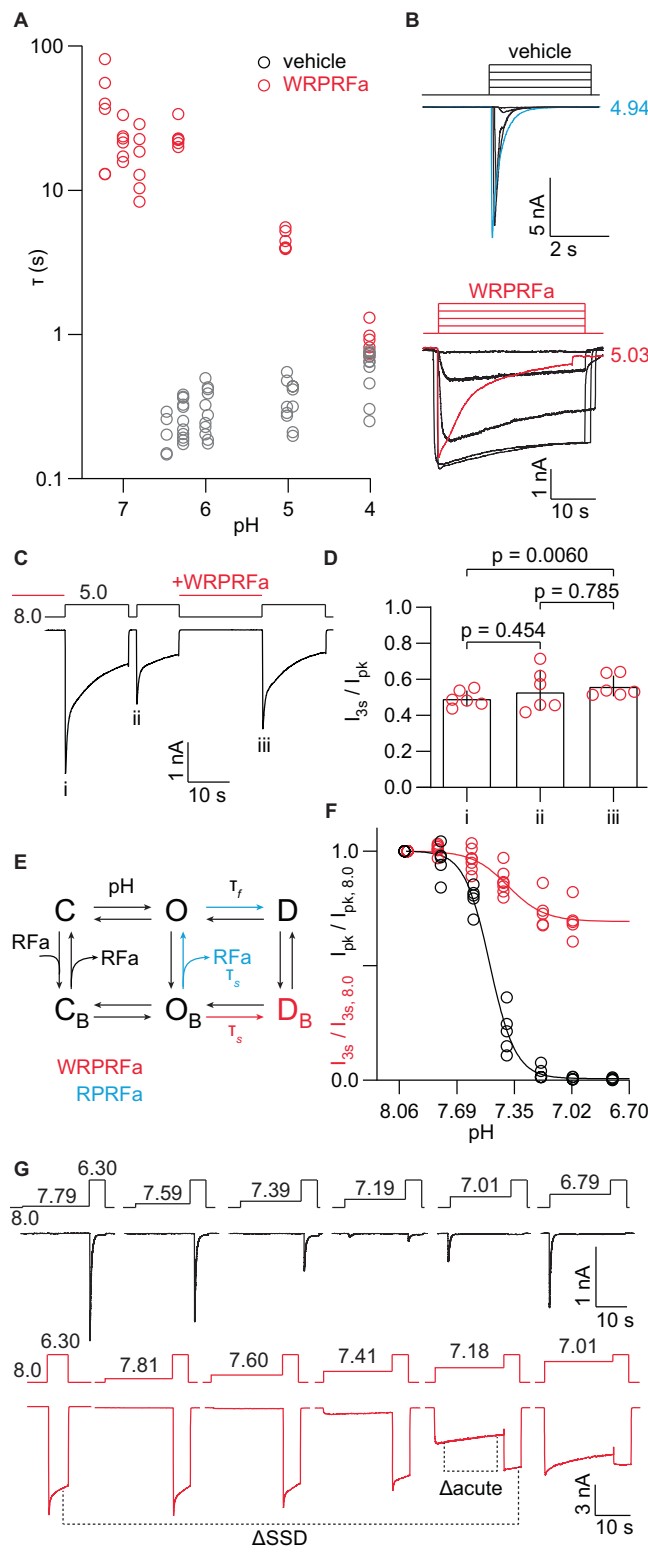

**Fig. 7 | WRPRFa slows ASIC3 desensitization in a pH-dependent manner. A** pH-dependence of desensitization rate. Traces were fit with a monoexponential decay to determine the time constant, or τ. Gray indicates rates measured in the absence of WRPRFa and red after treatment with 30 µM WRPRFa. Vehicle data are replotted from Fig. 6B. Data were collected from six cells. **B** Traces depicting measures plotted in (**A**) from single cells. Responses at nominal pH 5.0 are indicated for vehicle (blue) and 30 µM WRPRFa (red). **C** Protocol (top) and trace (below) for peptide unbinding experiment. Roman numerals indicate different peaks used to calculate sustained current fraction shown in (**D**). Cells were activated by pH 5.0 for 15 s (peak i), followed by a 2 s recovery at pH 8.0, then another 10 s at pH 5.0 (peak ii). Cells were recovered at pH 8.0 for 22 s, then activated at pH 5.0 again for 15 s (peak iii). **D** $I_{3s}/I_{pk}$ for each peak of the experiment shown in (**C**). Significance calculated by Tukey's multiple comparisons test following one-way ANOVA. **E** Gating scheme detailing the closed (C), open (O), and desensitized (D) states, with and without the presence of an RFamide[45]. T represents fast and slow decay components. Blue indicates the origin of the slow component for RPRFa-bound channels and red the slow component for WRPRFa-bound channels. **F** pH-dependence of steady-state desensitization measured as the fraction of current remaining during a pH 6.3 test pulse following a pre-conditioning hold at sub-activating pH, normalized to the current following a conditioning at pH 8.0. Black shows responses under vehicle conditions and red after treatment with 30 µM WRPRFa. Current from vehicle-treated channels is reported as $I_{pk}$ and from peptide-treated channels as $I_{3s}$. Data collected from five cells for vehicle-treated and seven cells for WRPRFa-treated. **G** Traces of the data plotted in (**F**). Δacute indicates current decrease from acute desensitization and ΔSSD the measurement of steady-state desensitization.

current after 20 stimulations (Fig. 8A and B, S10). At pH 5.0 only a small effect was seen. Preincubation with 30 µM WRPRFa had no impact at pH 5.0, but there was a marked acceleration in the tachyphylaxis observed at pH 4.0 (vehicle τ: 5.75 stimulations, 95% CI: 5.15–6.47 stimulations; WRPRFa τ: 1.39 stimulations, 95% CI: 1.26–1.52 stimulations; >99.9% τ are different).

ASIC1a tachyphylaxis only affects activated channels, with desensitization playing no recognized role. We tested if the same was true for ASIC3. Cells were held at pH 3.0, 4.0, or 5.0 for increasing durations (3 to 60 s) in a "static" protocol. A first test pulse to pH 6.3 established the baseline current, which was followed by the pH 3.0 - pH 5.0 conditioning step of varying durations. After the conditioning step was a 10 s recovery interval at pH 8.0, then another pH 6.3 test pulse that was normalized to the baseline test pulse to determine the fraction of channels lost during conditioning.

The results show a clear time- and pH-dependence where loss of current is faster under more acidic conditions (Fig. 8C and D) despite channels undergoing only a single activation at the conditioning pH. Like the pulsed protocol, the effect at pH 4.0 was enhanced by pre-treatment with 30 µM WRPRFa. Unlike ASIC1a, ASIC3 can undergo tachyphylaxis from the desensitized state as well as the activated state.

## Discussion

The results presented here describe the discovery and characterization of the RFamide, WRPRFa, that we used to answer questions about RFamide SAR and the mechanics of ASIC3 gating. WRPRFa is at least 4-fold more potent and has twice the maximal increase of sustained current on ASIC3 than RPRFa, which afforded us a greater signal window to detect small changes in RFamide SAR. We profiled key structural factors important for RFamide activity and established a glutamate pair in the lower palm domain interacts with the RFamide core arginine (site 2). We used its pronounced ability to slow ASIC3 desensitization to provide insights into the mechanisms of ASIC3 desensitization. Additionally, we demonstrated clear tachyphylaxis for ASIC3.

The greatest changes to activity were seen in the core RFa motif. Deamidation of the C-terminus in RPRF-COOH and WRPRF-COOH decreased their activity, as seen for other RFamides against ASICs[30,44]. At site 2, the strong preference for Arg over Lys indicates Arg is necessary for more than its positive charge; instead, the bulkier guanidinium group may be needed for coordinating multiple hydrogen bonds. Methylation of the guanidinium nitrogens progressively reduced the activity of site 2-modified

We inquired whether ASIC3 could exhibit tachyphylaxis and whether prior studies had employed insufficiently acidic solutions to detect it. HEK cells stably expressing ASIC3 were activated with pH 4.0 or 5.0 for 3 s at 10 s intervals for 20 pulses in a "pulsed" protocol. After activation by pH 5.0 or 6.3 WT ASIC3 recovers from desensitization completely in less than 10 s (Fig. S9A), with 90% of the current restored within 3 s for pH 6.3 and 6 s for pH 5.0. However, at pH 4.0 ASIC3 demonstrated a pronounced tachyphylaxis, with complete and irrecoverable loss of

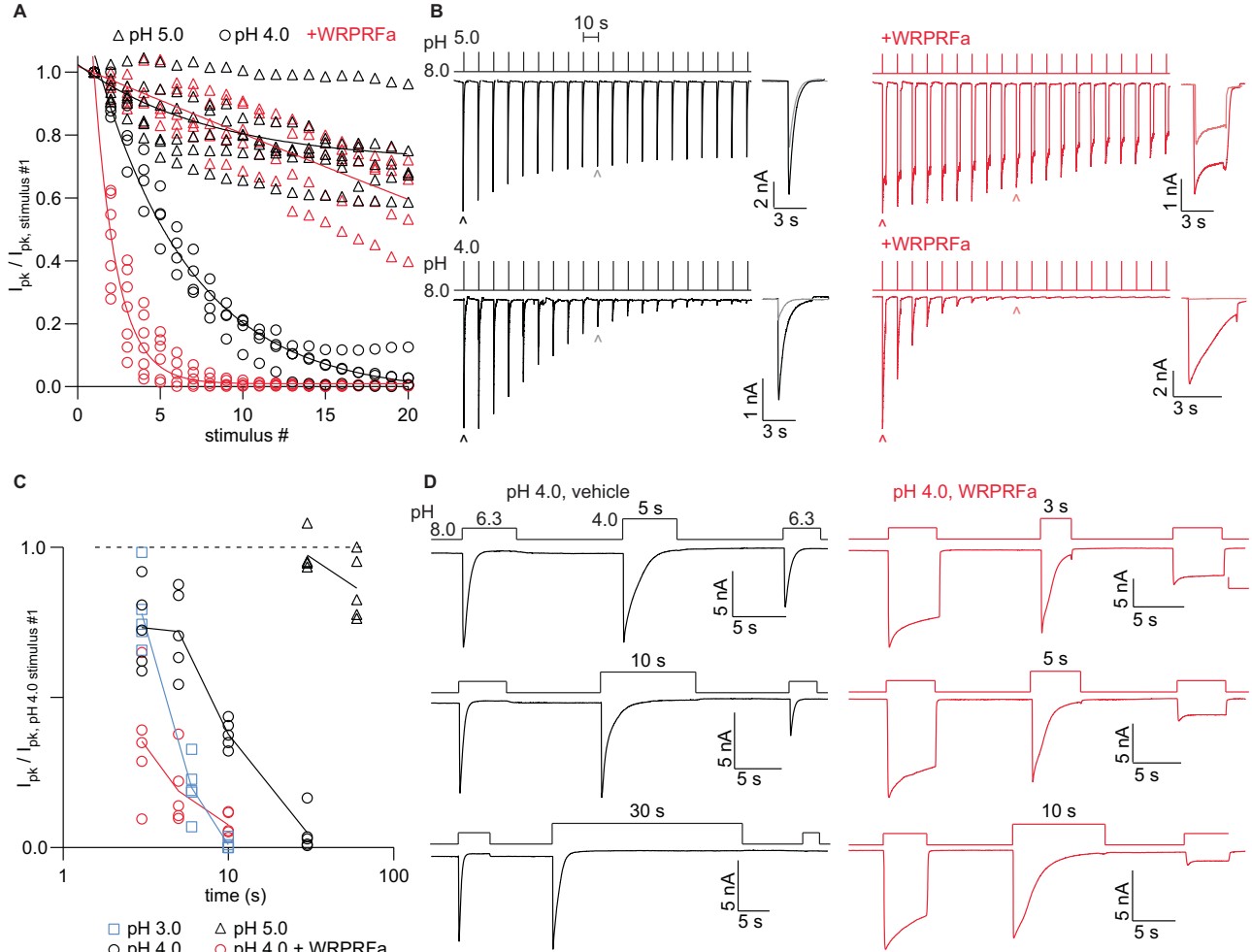

**Fig. 8 | WRPRFa accelerates ASIC3 tachyphylaxis. A** Decrease in ASIC3 current resulting from repetitive acid stimulations. pH 4.0 stimulus in circles and pH 5.0 triangles. Vehicle conditions are black and 30 µM WRPRFa-treated in red. Solid lines represent fits with a monoexponential decay function. Data collected from six cells for pH 5.0 with vehicle, five cells for pH 5.0 with WRPRFa, four cells for pH 4.0 with vehicle, and seven cells for pH 4.0 with WRPRFa. **B** Traces depicting data presented in (**A**). Expanded traces at right shows pulse #1 in dark lines and pulse #10 in light line, indicated with carets. **C** Current loss during hold at pH 3.0 (squares), pH 4.0 (circles), and pH 5.0 (triangles). Cells treated with 30 µM WRPRFa shown in red. Data collected from 20 cells for pH 4.0, 15 cells for pH 3.0, 10 cells for pH 5.0, and 15 cells for pH 4.0 with WRPRFa. **D** Representative traces of data shown in (**C**). Above, liquid protocol used in (**C**). Below, traces of ASIC3 current at indicated pH. Protocol in red indicates entire experiment was done in presence of 30 µM WRPRFa.

peptides, perhaps by disrupting hydrogen bond formation with side chain carboxylate oxygens of the E78/E421 pair. Between the dimethylated arginines, the asymmetric dimethyl (ADMA) showed a greater reduction of activity than the symmetric dimethyl (SDMA), which suggests both primary guanidinium nitrogens are required to participate in hydrogen bonding for maximum peptide activity. The arginine side chain pKa is not affected by methylation, therefore it is unlikely the decrease in activity is due to a change in ionization[58]. In contrast to our results, replacement of the RFa core Arg with Lys or Cit had no impact on either FaNaC or FaNaC1[40], whereas the activity of WRPKFa or WRP{Cit}Fa on ASIC3 was greatly reduced, indicating that FaNaC interactions with RFamides are fundamentally different from those we report here. Conversely, the replacement of Arg with Lys or Ile at site 4 showed a less abrupt decrease in activity. The contribution of site 4 may be a combination of cationic character and side chain length.

Sites 1 and 5, which otherwise favored large, cyclic side chains, were intolerant of His. The pH 6.3 stimulus used for activation is near the pKa of the His imidazole side chain (~6.0) and a partial positive charge may be incompatible with hydrophobic pockets. HRPRFa or WRPRHa could be more effective under alkaline conditions. While sites 1 and 5 both displayed hydrophobic preferences, the SAR indicates a subtle difference between the two. Replacement of Phe at site 1 with the bulkier pentafluorinated Phe had

little impact, and potency increased upon dearomatization of the phenyl ring in WRPR{Cha}a. In site 5, however, activity was reduced by half when Trp was replaced with Cha or Phe (F5). Cpg and Cpa, with their smaller cyclopentane and cyclopropane rings, respectively, were also less effective. The saturated rings are puckered and occupy more space than the flat, aromatic side chain of Phe. Thus, the binding pocket for site 5 may best suit a flat aromatic ring for π-stacking interactions, while the site 1 pocket is larger and accommodates bulkier groups.

Site 3 may be important partly as a spacer to position the flanking cationic/hydrophobic moieties, made evident by the activity loss going from FRPRFa to FRRRFa. Structural constraint at site 3 was not necessary; replacement of Pro by Ser or Ala caused only a modest reduction of activity. Like Pro, Tic is also a structurally constrained residue but with a larger side chain. Curiously, while site 3 Tic increased R{Tic}RFa activity, it had the opposite effect in WR{Tic}RFa, perhaps by interfering with the improved binding pose induced by the site 5 Trp.

The pharmacophore for RFamides in ASICs may be best described as a flat, aromatic hydrophobic moiety at the N-terminus and a bulky hydrophobic moiety at the C-terminus that positions two cations, one of which must form multiple hydrogen bonds (Fig. S11). Our screen tested only two concentrations, so it is difficult to distinguish whether changes

in activity arose from altered affinity for ASIC3 and/or attenuated functional effects. Furthermore, peptide affinity may change throughout the gating cycle. Thus, we cannot fully separate each residue's distinct contribution to affinity or efficacy.

Experiments to assess state-dependent binding indicate that WRPRFa preferentially interacts with the resting/closed state over the activated and desensitized states, yet WRPRFa dissociates from activated channels much more slowly than resting state channels. While binding to the activated state appeared negligible, the window of time for WRPRFa to bind the activated state before desensitization may be too short if the activated state on-rate is like that of the closed state. ASICs undergo structural rearrangement during desensitization[44,57] that may conceal or reconfigure the RFamide binding site, preventing binding in both the activated state and desensitized state protocols. Alternatively, the RFamide binding site in the activated channel could be "shut"; this would prevent WRPRFa from reaching its binding site as well as slowing its dissociation from the channel. High-resolution structures of ASIC3 bound to WRPRFa under neutral and acidic conditions would be useful to interpret these results.

Studies on marine invertebrate channel gating by RFamides have proposed binding of one (HyNaC[51]) or three (FaNaC[8]) ligands to a channel, although these bind at sites distinct from the lower palm domain in ASICs. Molecular docking models have indicated 1:1 binding stoichiometries for FRRFa and RPRFa with ASICs as well as simultaneous engagement of residues on multiple subunits. We constructed heteromeric channels as concatemers to titrate available binding sites. Each exchange of an ASIC3 subunit for a WRPRFa-insensitive ASIC1a subunit reduced the potency and activity of WRPRFa. Despite the activity loss observed for ASIC1[31], WRPRFa remained able to slow desensitization and induced an alkaline shift in the $pH_{50}$ of activation. From this, we expect a single ASIC3 subunit to provide a sufficient WRPRFa binding site and therefore only a single ASIC3 subunit is required for WRPRFa to exert its effect on desensitization. This is consistent with our observation that only one ASIC3 subunit is required to imbue the heteromeric channel with accelerated acute desensitization. However, maximal activity of WRPRFa likely requires a full binding site formed by three ASIC3 subunits.

Our double mutant cycle analysis of glutamates in the lower palm domain confirmed a functional coupling between the WRPRFa site 2 and a pH-sensing glutamate pair in ASIC3 (E78 and E421). This interaction was anticipated but had not yet been experimentally demonstrated[45]. The E78Q and E421Q mutations had opposing effects on WRPRFa activity; neutralization of E78 increased the potency of WRPRFa while neutralization of the E421 charge reduced its potency. In the chicken ASIC1a crystal structure shown in Fig. 4B, the carboxyl side chains of E78 and E421 are in close proximity (3-4 Å)[6,48], where they are believed to form a carboxyl-carboxylate pair with one of the side chains being protonated. In such pairs, the pKa of one glutamate, ordinarily ~4.0[59], may decrease 1-2 units and the other increase 2-3 units[6,60]. The guanidinium group of the site 2 Arg is expected to simultaneously engage the E78 and E421 side chains in a trinary complex. Mutation of either glutamate residue to glutamine would disturb the pKa shift of the intact glutamate, altering its interaction with arginine, which could explain the disparate effects of glutamine mutation on WRPRFa potency for E78 and E421. Disturbance of the pKa shift would alter the pH-dependence of side chain charge, consistent with changes in channel pH-sensitivity previously reported for mutations of the lower glutamate pair in ASIC1a and ASIC3[61,62]. Additionally, we found that the mutation to glutamine of the upper glutamates E378 and E416 did not affect WRPRFa activity, in contrast to the reported interaction of FRRFa and RPRFa with E374 (ASIC1a)[43] and E418 (ASIC3)[45], respectively. Alternatively, differences between the respective interactions of RPRFa and WRPRFa with rat and human ASIC3 could arise from species differences affecting the binding pocket.

In the absence of an ASIC3/RFamide structure, identification of this interaction provides valuable experimental evidence positioning WRPRFa within the ASIC3 central vestibule. This position would place the peptide adjacent to the β11-12 linker, which is necessary for both acute

desensitization and SSD. Isomerization of the EALN motif, or molecular clutch, within the β11-12 linker has been identified as a required process for channel desensitization, and modifications that restrict its movement affect the rate of desensitization[11,63]. WRPRFa may thus slow the acute desensitization of ASIC3 by sterically hindering movement of the molecular clutch or by modifying local electrostatics to shift the relative stability of activated and desensitized states. Details of the specific molecular mechanism by which WRPRFa modulates ASIC3 desensitization will require further study and be facilitated by co-crystal structures of ASIC3 with WRPRFa. Taken together, these results reflect the diversity of RFamide binding modes on the ENaC/DEG/ASIC family.

Acute desensitization is a pH-independent process that rapidly terminates channel activation, but after treatment with WRPRFa, desensitization is slowed and exhibits a clear pH-dependence. A prior study demonstrated RPRFa dissociates from ASIC3 before desensitization[45], and we hypothesized the increased rate of desensitization could be from charge neutralization of sidechains involved in binding. This would reduce the strength of binding WRPRFa under progressively acidic conditions, resulting in dissociation and allowing the channel to desensitize. Using a similar protocol, we did not detect an increased fraction of unbound channels, suggesting the lost current was due to bound channels entering the desensitized-bound state. To reconcile our findings with those published before[45], we suggest the acidic stimulus that elicits desensitization neutralizes the charge on E78/E421, weakening the interaction with the site 2 Arg. Interrupting this interaction permits the β11-12 linker to isomerize and desensitize the channel. RPRFa, which exhibits a weaker affinity to ASIC3, unbinds from the desensitized channel while WRPRFa remains attached due to the site 5 Trp. WRPRFa can then stabilize the desensitized-bound state and, furthermore, slow recovery to the resting state.

We show here that ASIC3 can undergo tachyphylaxis at pH 4.0. Previous studies that reported the absence of ASIC3 tachyphylaxis used stimuli of pH 5.0-5.6[12,57], consistent with our observation of minimal tachyphylaxis at pH 5.0. One explanation for the absence of tachyphylaxis in prior studies could be the use of insufficiently acidic stimuli; we observed ASIC3 tachyphylaxis only at pH 4.0, suggesting that a threshold concentration of protons may be required for tachyphylaxis to occur. Here, we also used a higher stimulation frequency than prior studies. Additionally, both prior studies used *Xenopus* oocytes for expression of ASIC3 in contrast to the HEK cells we used in this study. Differences in membrane composition or intracellular conditions may influence the extent and pH sensitivity of tachyphylaxis.

The rate at which ASIC1a undergoes tachyphylaxis is dependent on the stimulus pH as well as the time spent in the open state, but not the desensitized state. Contrarily, ASIC3 displays a similar loss of current from both the activated/open (Fig. 8A) and desensitized states (Fig. 8C). In both cases, the rate of tachyphylaxis in ASIC3 is pH-dependent and directly related to time at an acidic pH. This discrepancy raises the question of whether ASIC1a and ASIC3 are undergoing the same or distinct processes.

How does WRPRFa accelerate the loss of current under both pulsed and static protocols? One explanation is that slowed desensitization by WRPRFa prolongs time in the open state at a low pH under both protocols, increasing the time for the process underlying tachyphylaxis to occur. In this way, WRPRFa promotes an ASIC1a-type tachyphylaxis. Alternatively, the results presented in Fig. 7C and D suggest the WRPRFa-bound desensitized ASIC3 may have a slowed recovery from desensitization; thus, the accelerated tachyphylaxis in the presence of WRPRFa could be a combination of both tachyphylaxis and accumulation of WRPRFa-bound channels in the desensitized-bound state. Finally, tachyphylaxis refers to a decrease in activity after repeated doses or stimuli, which is inconsistent with our observations that ASIC3 currents can be lost irrevocably during a single activation. Future studies exploring these mechanisms may yield more appropriate terminology.

Overall, our results show that WRPRFa is an effective tool to probe the biophysics and biology of ASIC3. The SAR we established for WRPRFa will be useful for design and interpretation of ASIC3 structural and gating studies. Furthermore, we confirmed a predicted ASIC-RFamide interaction and demonstrated the presence of tachyphylaxis in ASIC3.

## Methods

### Chemicals
All chemicals were purchased from Sigma-Aldrich (St. Louis, MO).

### Peptides
Custom peptides were purchased from Genscript (Piscataway, NJ) with purity >95%. Peptides were dissolved in pH 8.0 extracellular recording buffer to make 30 mM stock solutions except WRPIFa and WRPR{Cpg}a, which were dissolved in DMSO to make 30 mM stock solutions. All chiral amino acids were L-configuration.

### Molecular biology and channel constructs
cDNA clones for human ASIC1a (NM_001095.4), human ASIC2a (NM_001094.5), and human ASIC3 (NM_004769.4) were cloned into a pcDNA3.1-P2A-eGFP (ASIC1a, ASIC2a) or pcDNA3.1-P2A-mCherry (ASIC3) mammalian expression vector (Thermo). The P2A element self-cleaves during translation to separate the C-terminal fluorescent reporter from the channel.

ASIC131 and ASIC313 constructs were created by placing human ASIC1a and human ASIC3 ORFs in tandem in the pcDNA3.1-P2A-mCherry vector, with the first and second ORF attached by a AGSVDS linker sequence and the second and third ORF by an AGSGSS linker.

All mutagenesis was performed at Genscript, and sequences were verified by nanopore sequencing by Plasmidsaurus (Eugene, OR). The numbering of ASIC3 amino acids is based on RefSeq NP_004760.1. Plasmids were transfected into CHO cells using Lipofectamine 3000 (Thermo) according to the manufacturer's protocol. Cells were used for patch-clamp recordings 24–48 h after transfection.

### Sequence alignment and structural analysis
DNA and protein sequences were aligned using Clustal Omega (EMBL-EIB, Cambridgeshire, UK). A published structure of chicken ASIC1a (PDB: 6AVE) was used to generate structural models with UCSF ChimeraX (version 1.10).

### Mammalian cell preparation
CHO cells stably expressing human ASIC1a (NM_001095.2) (CT6012; Charles River Laboratories) were grown in Ham's F-12 (Life Technologies), 10% FBS, tetracycline free (Sigma), and 1% penicillin-streptomycin (Thermo) with 10 μg/ml Blasticidin (Thermo) and 400 μg/ml Zeocin included as selection antibiotics. Media was exchanged to remove selection antibiotics and cells were induced with 2 μg/ml tetracycline the morning of recording (2–6 h prior to start of the experiment).

CHO cells stably expressing human ASIC3 (NM_004769.1) (CT6173; Charles River Laboratories) were grown in F12K medium (Thermo), 10% FBS, tetracycline free (Sigma), and 1% penicillin-streptomycin (Thermo) with 10 μg/ml Blasticidin (Thermo) and 400 μg/ml Zeocin included as selection antibiotics. Media was exchanged to remove selection antibiotics and cells were induced with 2 μg/ml tetracycline the day before recording (15-18 hours prior to start of the experiment).

HEK cells stably expressing human ASIC3 (NM_004769) (CYL3055; Eurofins DiscoverX) were maintained in media containing DMEM/F-12 (with L-Glutamine), 10% FBS, 1% non-essential amino acids (Thermo-Fisher), 1% penicillin-streptomycin, and 400 μg/mL Geneticin as a selection agent.

CHO cells (85050302, Sigma) used for transfection were maintained in media containing DMEM + Glutamax, 1% Glutamax, 10% FBS, 1% NEAA, and 1% penicillin-streptomycin.

All cells were cultured at 37 °C with 5% $CO_2$. Cells were lifted with TrypLE dissociation reagent (Thermo) or Detachin (Amsbio, Cambridge, MA) and plated onto uncoated German glass coverslips (Neuvitro) with antibiotic-free media. Cells were thereafter kept at 30 °C with 5% $CO_2$ and used 1–2 days after plating. For mutants with low expression, 5 mM sodium butyrate was added to the culture media the day before recording to increase current size.

All cell lines were confirmed negative for mycoplasma contamination with the MycoAlert PLUS Detection Kit (Lonza).

### qRT-PCR expression quantification
HEK cells stably expressing ASIC3 were lifted with Detachin and washed once with DPBS without $Mg^{2+}$ or $Ca^{2+}$, then centrifuged for 10 min at 500 x $g$. RNA was harvested from the cell pellet with an RNeasy Mini Kit (Qiagen, Germantown, MD). An EXPRESS One-Step Superscript qRT-PCR Kit (ThermoFisher) used to setup the reverse-transcription and amplification reactions. The experiment was run on a StepOnePlus Real-Time PCR system (Applied Biosystems, Singapore) and analyzed with StepOne Software v2.3 (Life Technologies). Standard curves for ASIC1a and ASIC3 were made by 1:10 serial dilutions of human ASIC1a (NM_001095.4) and human ASIC3 (NM_004769.4) clones in a pcDNA3.1-P2A vector backbone. Primer-probe assays for ASIC1 (Hs.PT.56a.1262264; exon 10-11) and ASIC3 (Hs.PT.58.24784572; exon 5–7) were purchased from IDT (San Diego, CA).

### Electrophysiology – MPC
CHO and HEK cells were tested in the whole-cell configuration on MPC platforms using an Axopatch 200B amplifier (Axon Instruments, San Jose, CA) and a Digidata 1322A or Digidata 1440A digitizer (Axon) using pClamp 9.2 or 10.7 (Molecular Devices, San Jose, CA), respectively. When testing transfected cells, red fluorescence from the cleaved mCherry reporter was used to identify cells expressing ASIC3, ASIC131, or ASIC313. For cells expressing transfected ASIC1a or ASIC2a, the green signal from the cleaved eGFP reporter was used. All electrophysiology data were analyzed with Clampfit 11.1 (Molecular Devices).

All recordings were done at room temperature (21 °C–23 °C). A low-volume (~300 μL) RC-26GLP recording chamber (Warner Instruments, Hamden, CT) was used with two gravity perfusion systems. Peptides used for pre-treatment were applied at pH 8.0 using a bath perfusion system with a flow rate of ~8 mL/min and flowing for at least 1 min. All other solutions were focally applied by a Millimanifold perfusion pencil (ALA Scientific, Farmingdale, NY) with a flow rate of ~3 mL/min. The patch-clamped cell was positioned in the outlet of the perfusion pencil, and solution was flowed continuously throughout the experiment to maintain the applied solution and pH around the cell. The focal perfusion manifold was digitally controlled with a BPS-8 (ALA Scientific) or VM8 (ALA Scientific) electronic controller. Channels began to activate about 100 ms after the electronic valve signal.

The external bath solution was made of 135 mM NaCl, 5 mM KCl, 1 mM $MgCl_2$, 1 mM $CaCl_2$, 10 mM HEPES, 10 mM 4-morpholineethanesulfonic acid (MES), 10 mM sodium acetate, and 5 mM D-glucose, adjusted to a nominal pH with NaOH or HCl. Bath solution pH was measured with an Orion STAR A series pH meter (Thermo) before recording, and the measured pH was reported. The pipette solution contained 65 mM CsF, 65 mM CsCl, 10 mM HEPES, 5 mM EGTA, 1 mM $MgCl_2$, adjusted to pH 7.4 with CsOH. Mannitol was used to adjust osmolarity, with the pipette solution kept 10-15 mOsm hypoosmotic relative to the bath solution, measured with a 5004 MicroOsmette Osmometer (Precision Systems, Natick, MA). Peptides were diluted directly into the external bath solution except WRPIFa and WRPR{Cpg}a, which were diluted 1000-fold to give 0.1% DMSO in the final solution. Addition of peptides up to and including 100 μM did not affect the bath solution pH.

Pipettes were pulled from borosilicate glass (Sutter Instruments, Novato, CA) with a P-97 Flaming/Brown Puller (Sutter) and polished on a

MF-9 Microforge (Narishige, Tokyo, Japan). When filled with intracellular solution, pipette resistances were 1.5–4 MΩ. Series resistance was <10 MΩ and uncompensated. Signals were low-pass filtered at 1 kHz and digitized at 10 kHz for voltage-clamp experiments. Experiments used -60 mV as the holding potential and pH 8.0 as the holding pH. Liquid application protocols are visually described in the figures. All electrophysiology experiments were performed by MPC as a default unless otherwise specified.

## Electrophysiology – Qpatch II

RFamide peptide activity on ASIC3 was measured on the QPatch II platform (Sophion, Ballerup, Denmark) using either a CHO or HEK stable cell line and analyzed with Sophion Analyzer software (version 5.2). Prior to recording, the cells were washed twice in PBS (Gibco, Paisley, UK) and treated with Detachin (Genlantis, San Diego) for 3 min, then resuspended in their respective growth media without serum for a concentration of $2 \times 10^6$ cells/ml. A minimum of 5 mL of cells was added to the QStirrer, which was placed on the deck of the QPatch II and allowed to incubate at room temperature for a minimum of 20 min.

The bath solution contained 135 mM NaCl, 5 mM KCl, 4.8 mM $CaCl_2$, 1 mM $MgCl_2$, 10 mM HEPES, 10 mM MES, and 10 mM glucose adjusted to pH 7.4 with NaOH. The intracellular solution contained 65 mM CsF, 65 mM CsCl, 10 mM HEPES, 5 mM EGTA, 1 mM $MgCl_2$, adjusted to pH 7.4 with CsOH. Cells were held at -60 mV and ASIC currents were evoked with 3 s applications of pH 6.3 buffer. Peptides were preincubated for at least 2 min. The extracellular solution was changed by dispensing 5 µL of buffer through the ~1.7 µL recording chamber, with a solution switch time of 100 ms, that exchanges 99% of the solution. Signals were acquired at 1 kHz and low-pass filtered at 300 Hz. Experiments performed on QPatch II are specifically indicated in the text.

## Statistics and analysis

qPCR results were analyzed by fitting the standard for ASIC1a and ASIC3 with the semilog equation:

$$Y = Y_{int} + \log(X) * slope$$

where Y is the Ct value and X the gene copy number. ASIC1a and ASIC3 Ct values from the ASIC3 HEK stable cell RNA were interpolated with the respective standard curve to yield a copy number for each gene.

Electrophysiology recordings were analyzed as indicated in prior sections. GraphPad Prism 10.3.1 was used to analyze results, perform fits, and generate plots.

Concentration-response data for peptides as well as the relationships between activation and pH, and desensitization and pH, were fit with the non-linear regression: "[Agonist] vs response – variable slope (four parameters)" using the equation:

$$Y = Bottom + \frac{Top - Bottom}{1 + \left(\frac{IC50}{X}\right)^{Hill\ slope}}$$

For peptide concentration-response curves, Bottom was set to 0 and Top was constrained to ≤ 1 for peptides that approached plateau phase or where $I_{3s}/I_{pk}$ reached 1. For peptides that did not exhibit a plateau, we constrained Top equal to the geometric mean of the highest concentration and reported the $EC_{50}$ as a lower limit of the true value. For activation- or desensitization-pH curves, Bottom was set to 0 and Top constrained to 1. Concentration-response data reported an $EC_{50}$ and 95% confidence interval (CI). pH-dependence of activation or desensitization reported a $pH_{50}$ and a 95% confidence interval.

Monoexponential fits used the non-linear regression: "one phase decay" using the equation:

$$Y = (Y0 - Plateau) * \exp(-K * X) + Plateau$$

where K is the rate constant and its reciprocal τ, the time constant, was reported with a 95% confidence interval. For fits to activated-state and resting-state wash off, $Y_0$ was set to 1.

Coupling coefficients (Ω) were calculated using the equation:

$$\Omega = \frac{(WT|WRPRFa) * (E78Q|RFa')}{(E78Q|WRPRFa) * (WT|RFa')}$$

where each parenthetical represents the $EC_{50}$ of the given combination of channel (WT ASIC3 or E78Q) and RFamide (WRPRFa or RFa', a modified peptide)[49].

Results from the QPatch II SAR screen were compared by ordinary one-way ANOVA and pairwise comparisons were made with Dunnett's multiple comparisons test. The family-wise α was set at 0.05, and multiplicity-adjusted p-values are reported in the text and tables. The significance threshold for other comparisons (i.e., t-test) was set at 0.05. Tests of statistical significance are indicated in the main text or corresponding figure legend.

Data points from multiple cells were combined into a single plot before curve fitting. For pH-dependence of activation or time-dependence experiments, data from each cell were normalized to an indicated stimulus before addition to the combined plot. For the peptide concentration-dependence of activity, the $I_{3s}/I_{pk}$ value was calculated before addition to the combined plot. The number of cells tested for each condition in a plot is indicated in the figure legend.

All curves represent global fits to the plotted data, and values derived from fitting approaches are reported in the text with 95% confidence intervals. Parameters for global curve fits (i.e., $pH_{50}$, $EC_{50}$, τ) were compared with Akaike's Information Criterion (corrected), and we reported the probability that the indicated parameter was different between the compared fits[64].

Individual data points were plotted except when doing so would obscure the presentation of the data. In cases where averages are plotted, points represent geometric means and error bars geometric standard deviation (geometric mean ×/÷ geometric standard deviation). All averages stated in the text or tables are geometric means with geometric standard deviations, except when the data set contained negative values, and we instead report the arithmetic mean with standard deviation (arithmetic mean ± standard deviation). Such instances are noted where they occur. Measures for a single condition (pH, concentration) were taken from multiple samples. This study was not designed to test a prespecified statistical null hypothesis and a prewritten statistical protocol was not used.

## Reporting summary

Further information on research design is available in the Nature Portfolio Reporting Summary linked to this article.

## Data availability

All data supporting the findings of this study are available within the paper and Supplementary Information. Source data used to generate figures is provided in Supplementary Data 1.

## Code availability

No custom code or software was used for the collection or analysis of data.

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

## Acknowledgements
We would like to thank Bryan Moyer, Deborah Dawson, and Ya-Wen Lu for critical reading of the manuscript, Victor Castro for performing qPCR experiments, Tina Holt for registering peptides, and Arthur Gilchrist for insightful commentary. No specific funding was received for this work. This work was fully supported by Latigo Biotherapeutics, Inc.

## Author contributions
Conceptualization: C.C., N.D.Y., V.J., and J.M.G. Investigation: C.C., N.D.Y., V.J., S.M.A., and J.M.G. Methodology: C.C., N.D.Y., V.J., and J.M.G. Supervision: J.M.G. Visualization: J.M.G. Writing – original draft: J.M.G. Writing – reviewing and editing: C.C., N.D.Y., V.J., and J.M.G.

## Competing interests
All authors were employees of Latigo Biotherapeutics, Inc at the time the work was performed.
