## [Transparent Peer Review file · Communications Chemistry]

Identification, characterization, and structure-activity relationship of the ASIC3-selective peptide WRPRFa

Corresponding Author: Dr John Gilchrist

Version 0:

Reviewer comments:

Reviewer #1

(Remarks to the Author)

In this manuscript, Chien et al. report the discovery of a new RFamide peptide, WRPRFa, that modulates acid-sensing ion channel 3 (ASIC3) more strongly than previously described RFamides. They go on to replace systematically individual WRPRFa residues by several other natural and unnatural amino acids to determine the structure-activity relationship. Moreover, they performed experiments to determine the state-dependence of WPRFa binding to ASIC3. In addition, they replaced individual amino acids on ASIC3 to identify a possible binding site and used double mutant cycle analysis to confirm the interaction of an Arg residue of WRPRFa with an acidic residue of ASIC3. Then they used concatameric ASIC1a-ASIC3 heteromers with a fixed number of ASIC3 and ASIC1a subunits to analyze the influence of the number of ASIC3 subunits on modulation by WRPRFa. In addition, they used WRPRFa modulation to shed light on two different schemes that had been used to explain ASIC gating. Finally, they report for the first time tachyphylaxis—a reduced response of the channel with repeated activation—for ASIC3.

This is an interesting manuscript that contains a lot of information. There is no doubt that WRPRFa much more strongly delays ASIC3 desensitization than previously described RFamides. However, not all conclusions are supported by the data, some claims are overstated, and not all findings are novel.

Main comments:

1) In the abstract, the authors claim that the RPRFa peptide “has an unresolved mechanism and binding site”. However, later in the manuscript, it becomes clear that a putative RPRFa binding site has previously been identified by molecular docking and site-directed mutagenesis. The binding site proposed in the present study involves similar amino acids and, thus, at least overlaps with the previously identified binding pocket. This is also correctly stated later in the abstract (“...and identify an interaction site overlapping with other RFamides”). In addition, a mechanism for slowed desensitization by RPRFa had also previously been proposed. These previous findings are largely confirmed in this study. Claims of novelty should therefore be toned down.

2) The authors used HEK and COS cells in their study. It was not always clear to me which cells were used for which experiments. This should be more clearly stated in the figure legends. HEK cells contain endogenous ASIC1a. Stably expressing ASIC3 in these cells should give rise to at least a subpopulation of ASIC1a-ASIC3 heteromers, which should contaminate measurements. How was a possible contamination by heteromers assessed? Likewise, it was not clear to me, in which experiments automated patch clamp was used and in which manual patch clamp. This should also be stated more clearly.

3) The authors provide a precise EC50 value for RPRFa (66.4 μ M), although the data points do not saturate and only allow to provide an estimate of apparent RPRFa affinity (Fig. 1B). Either measurements should be provided using more RPRFa concentrations or conclusions regarding differential affinities of RPRFa and WRPRFa should be toned down.

4) How well was the pH in the measurements controlled, in particular for the measurements using automated patch clamp? The authors should provide more details on this. Was there a constant solution exchange or was solution exchanged only once? If so, how much of the solution was exchanged? These details are relevant to judge the quality of the measurements. It was also not clear whether the pH of all solutions containing peptides had been adjusted after adding the peptides. Please state this explicitly.

How was the pH measured? In some measurements the authors indicate very precise pH values, for example in Fig. 6B

"4.94", when the "nominal pH" was 5.0. What does this mean?

5) Tachyphylaxis. It is surprising that the authors, for the first time, report tachyphylaxis for ASIC3. If I see it correctly, previous studies used pH 5.0 and pH 5.6 to assess tachyphylaxis of ASIC3, not 5.5-6.0 as stated in line 572. Why has tachyphylaxis not been noticed in these previous studies? Different expression systems as a possible source for this variability should be discussed. Is it possible that "tachyphylaxis" in the present study is due to a more classical "rundown"? Another potential issue is solution pH exchange. Was it complete? Or is it possible that the acidic pH to activate ASIC contaminated the conditioning pH? If so, experiments with a more complete solution exchange should be performed.

6) Stoichiometry: the authors largely confirm a WRPRFa binding site in the lower palm domain as previously proposed for RPRFa on ASIC3 and other RFamides on ASIC1a. It should be noted, however, that previous studies reported binding of RFA peptides in a single cavity at the center of the threefold symmetry of the channel. Molecular docking confirmed a single RPRFa molecule in this cavity. The results of the authors do not contradict this view. Therefore, either the authors should provide compelling evidence for the binding of three peptides simultaneously to ASIC3 (is there enough space to accommodate three peptides at once?) or they should change their interpretation. The interaction of E78 with site 2 of WRPRFa has already previously been proposed by docking results. This finding is therefore not completely novel but rather confirmatory. This should be mentioned.

7) For the interpretation of some measurements, it would be helpful to describe the action of WRPRFa more carefully. For example, it appears that at 30 μ M WRPRFa—the concentration used in most experiments (the concentration should be indicated on the figure)—not all channels had bound WRPRFa (Fig. 1I). I think the initial current decline arises from a mix of channels with no WRPRFa bound undergoing desensitization and of unbinding of WRPRFa from other channels. Moreover, it appears that at acidic pH, the peptide unbinds much more quickly (Fig. 3B). Thus, binding of WRPRFa to ASIC3 is incomplete and complex. This caveat should be kept in mind when interpreting experiments designed to shed light on gating mechanisms.

8) The authors used WRPRFa to investigate gating mechanisms of ASIC3. I could not follow all their arguments and found some interpretations not convincing.

First, they find that the pH₅₀ of activation strongly shifted to the left so that it became very similar to the pH₅₀ of steady-state desensitization (SSD). It appears to me that this is exactly what a linear gating scheme shown in Fig. 6C would predict for a substance that almost abolishes desensitization.

In lines 365-366, they state that "acute desensitization happens under more acidic conditions". This is incorrect. In the simplest case, acute desensitization happens with a fixed rate constant irrespective of pH. It may appear slower at higher pH because not all channels desensitize at the same time.

In lines 390-391, they state that SSD of WRPRFa-bound channels should not be possible under the linear model. It was not clear to me why it should be possible in a circular model. It appears that WRPRFa (largely) prevents desensitization. If there is a single "D" state in both models, why can it be reached by WRPRFa-bound channels in the circular model but not the linear model?

The authors attributed the SSD to the difference in I₃s between the conditioned test pulse and baseline (lines 416-418). How can they exclude that this difference is in part due to unbinding of the peptide at the more acidic pH? The interpretation of the whole set of these experiments is confusing and not convincing. In my opinion, SSD should be estimated from the decline of the peak current, because it arises (at least in part) from acute desensitization of channels that had no peptide bound. If the peptide prevents desensitization, there should be no SSD in both models. I doubt that this peptide can be used to convincingly differentiate between a linear and a circular model.

9) The experimental design for some experiments shown in Figure 3 is not optimal. In Fig. 3D, there is no time interval between peptide application in the conditioning period and during activation at pH 6.3. Thus, the measurement on the right of the trace is not "the same" (line 226) and not directly comparable to the one on the left of the same trace. In Fig. 3F and G, to determine the off-rate from resting channels, it would have been better to use different times of wash-out followed by a single activation (and not to regularly activate ASIC3, even not for 5 sec). Perhaps it is better to show the experiment to determine the off-rate from activated channels first, because it allows to estimate the error introduced by repetitively activating channels.

10) The paragraph starting with line 548 is highly speculative and should be deleted or strongly shortened. If this model is correct, I would expect that WRPRFa unbinds during activation because E421 is no longer charged.

Minor comments:

1) Line 50: "with the first transmembrane domains as the wrist". I think it should be the forearm. The wrist is between the transmembrane domains and the ECD.

2) At several instances, more references would aid the reader not so familiar with the literature. For example, line 86 (references for the binding site in the lower palm domain),

3) Line 94: "revealed distinctions". Please be more specific.

4) Lines 161 and 188: the authors underline the "ringed side chain" of a His residue. I think given that the imidazole group can be protonated, it is not so curious that His behaved different than other aromatic amino acids.

5) Lines 264-266: "we saw four glutamates in two pairs". The pairs cannot be clearly seen in Fig. 4B. Only later (in lines 291-292), the authors mention previous crystal structures that revealed these pairs. These studies should be mentioned and cited earlier to make the notion of acidic pairs more convincing.

6) Lines 326-328: There are now several published studies that used ASIC concatamers. Therefore, I recommend deleting this sentence, which is somewhat diminutive with respect to previous work by others. Simply state why you used concatamers.

7) Line 338: Delete "accelerated"

Reviewer #2

(Remarks to the Author)

This study investigates the function of a short peptide that selectively modulates ASIC3. The authors try different modifications of the known ASIC3 modulator RPRFa, and find that N-terminal addition of a Trp residue strongly increases the effect of the peptide, leading at 100 μ M to a completely non-desensitizing current. They test how exchanging residues at either of the five positions of WRPRFa affects modulatory properties of the peptide, identifying different, position-dependent sensitivities to such changes. Of the potential interacting residues in the binding site on ASIC3, mutations of acidic residues were tested, identifying E421 as important for the modulatory effect of the peptide. Experiments with concatameric constructs indicate that one WRPRFa-binding subunit in the trimer is sufficient for the peptide to exert full modulation of ASIC3 desensitization. Finally, the authors observe tachyphylaxis that is increased by WRPRFa, if they stimulate ASIC3 at high frequency with very acidic pH solutions. Overall, this is an interesting description of a new ASIC3-modifying peptide.

Major points

1. The analysis of the ASIC3 pH dependence showed a substantial alkaline shift in the presence of WRPRFa. In the peptide-free condition, the test pH corresponds to the pH50. For this reason, the exposure to the peptide should, based on the pH50 shift, lead to a 2-fold increase in the peak current. This is evident in some of the current traces shown, but not in others. A possible change in peak current upon peptide exposure is not measured quantitatively. Could the authors provide such an analysis in the manuscript and interpret the finding?

2. Methodological details: The description of some experiments is not clear enough, as listed below. These aspects should be clarified.

2.1. Line 105, Many subtypes of HEK cells contain endogenous ASIC1a. Indicate in the methods whether you can confirm that your cell line does not express ASIC1a.

2.2. Please indicate in the methods, which data were obtained with manual patch-clamp, and which with the Qpatch II.

2.3. For the experiments with QPatch II, indicate how the cells were prepared for the measurement, specify also the low-pass filtering applied.

2.4. Indicate how concentration-response curves (for peptide and pH) were obtained, whether full curves were obtained on single cells or whether a limited number of data points were measured on each cell. In case of the second variant, indicate how data points from different experiments were combined. In the figure legends, the number of full, independent experiments should be indicated. If concentration-response curves were pooled from different experiments, indicate also how statistical comparison between different conditions was done (for example pH50 with or w/o the peptide).

2.5. On the axis titles and figure legends make it clear, what is really plotted. "I/I8.0" is not sufficient. It needs to be indicated whether I is the peak current or the current after 3s. Provide the timing information, i.e. when exactly an amplitude was measured, in the legends.

3. Logics of mechanistic conclusions.

3.1. State dependence. Lines 209-224, the I_{3s}/I_{pk} ratio is compared between the protocols in which peptide is applied to the closed, open or desensitized channels (Fig. 3). Although this ratio is higher after exposure to the desensitized than to the open channel, the interpretation given suggest a more important binding to open than to desensitized channels. In comparison, the binding to the closed channel would be more important. Fig. 3F shows then that the peptide unbinds rapidly from closed channels, but not from open channels. Could the authors provide a conclusion that takes all these findings into account? The fact that association is quite slow plays probably an important role in the outcome of the experiments of Fig. 3.

3.2. The hypothesis presented on lines 548-563 is not consistent with measurements in several published studies, that mutations of E78 or E421 result in only small shifts in pH dependence. It should be revised. It was however observed in ASIC3, that the mutation E79A (corresponding to E78 in the present study) induces a strong alkaline shift in the pH dependence of SSD (PMID: 17389250). This might be a promising basis on a hypothesis on the mechanism by which this domain is involved in SSD.

4. Measurement of the pH dependence of activation and SSD. In some experiments (Fig. 3A, lines 204-208), the activation pH dependence on the peptide-exposed channels is measured not as the pH dependence of the peak current, but the pH dependence of the I_{3s}/I_{pk} ratio. In other experiments, as for the concatamers, the analysis seems to have been done on the peak current, but this is not completely clear. While I understand the reasoning behind it, this is not correct. It is ok to determine the pH dependence of the I_{3s}/I_{pk} ratio as an additional measurement, but this does not correspond to the pH dependence of the channel activation of channels exposed to the peptide. For this, the pH dependence of the peak value has to be reported. A similar analysis was done for the SSD, as described in the text (lines 400-407), although this is not completely clear in the corresponding Fig. 6E.

5. Discussion of possible binding sites and number of peptides that bind to one channel trimer. Line 316 ff, it is stated that

E78 and E421 are located away from subunit interfaces. However, these residues point towards the central cavity of the channel, and there is a clear possibility that a peptide may interact with more than one subunit. The comparison with binding of such peptides to FaNaC and HyNaC (lines 318-321) is misleading if it is not indicated that in these peptide-gated channels, the binding site is in a completely different channel domain. The experiments with concatemers show that maximal peptide effects in channels containing 1 or 2 ASIC3 subunits are as high as in ASIC3 trimers, but that the concentration dependence is shifted. Some of these points are taken up in the discussion (lines 508-518) and should be adapted. Towards an understanding of the number of peptides binding, could the authors check whether the structures could accommodate three WRPRFa peptides in a single trimer without clashes? This information would be useful for the interpretation.

6. Tachyphylaxis experiments. Tachyphylaxis was observed when the channels were activated with high frequency, every 20s. Tachyphylaxis consists in an irreversible loss of function, and it is stated in the text (line 438 and elsewhere) that the loss was irrecoverable. The corresponding data are however not shown. Is this really tachyphylaxis, or just accumulation of channels in the desensitized state? The authors have to test whether with longer recovery periods at pH8, some of the current can be recovered. In this context, the time course of recovery from desensitization is very important. This is shown in ASIC3 in the absence of peptide in Fig. S10. To understand the possible mechanism by which WRPRFa contributes to tachyphylaxis, it is essential to provide also data on the recovery in the presence of the peptide.

Specific comments

1. Line 43, BASIC, sometimes called ASIC5, does not belong to the ASIC subfamily.
2. Lines 43ff, "All isoform...", Homotrimers of ASIC2b or ASIC4 do not respond to extracellular acidification.
3. Line 58 "neutral pH", I guess you mean pH7.4. "Physiological pH" might therefore be more appropriate.
4. For several experiments, EC50 values are indicated, where no saturation of the pH dependence is observed. This may be ok if the I3s/lpk hits the maximum of 1, but for other experiments it is not justified, as in Fig. 1A, and some of the data in Figs. 4F-I, an EC50 value cannot be determined and can be at best indicated as lower limit of the EC50.
5. Line 113, the difference to the previous study might be due to ASIC3 from different species, rat vs. human.
6. line 167, "Ile and Asn were...", clarify this sentence and provide statistical support for the conclusion.
7. The kinetics of the peptide binding and unbinding (Fig. 3E-F) are important observations, and in my view, they are also required to better understand the state dependency protocols of Fig. 3C-D. It would seem to me more logical to place them before panels C and D.
8. line 310, what does "46.5% pH50 are different" mean?
9. The part about the two kinetic models (lines 408-423) is hard to follow. Consider rewriting it.
10. Line 432, the cited study investigated ASIC1a, not ASIC3
11. Overall, the discussion on lines 571-595 is very long and could be shortened.
12. In many figure legends, the concentration-response experiments are called "EC50 for...". Similarly, it is said "dotted lines depicts the RPRFa EC50...". This is not the EC50, but rather the concentration-response curve.
13. Fig. 1A, I am surprised about the kinetics and the amplitude of the ASIC2a current at pH6.3. Can the authors double-check that this is indeed ASIC2a, and that the pH was 6.3?
14. Fig. 6F, is "current decrease" relative to the preceding pH or relative to pH8?
15. Fig. 7C, what are the solid lines?
16. Overall, it would be good to increase the line thickness of the symbols in the figures, as in the present version, it is not easy to recognize the different colors.

Reviewer #3

(Remarks to the Author)

This manuscript by Chien and colleagues describes the pharmacological and biophysical characterisation of a synthetic peptide, WRPRFa, that alters desensitisation properties of ASIC3-containing channels. ASICs remain an active area of investigation given their links to sensory physiology, pain pathways, and other diseases involving acidosis. Developing ligands that selectively modify specific gating properties is valuable not only for dissecting ASIC function and mechanisms, but also for distinguishing contributions of different subunit assemblies. In this context, the study positions WRPRFa as a potentially useful pharmacological probe with implications for both basic research and future drug discovery.

Overall, this is a nice paper that would make a good contribution to the literature. Below are a few comments for the authors to consider, along with some minor points listed.

1) Fig 1A-C (Line 113): With the reported EC50 of 66 μ M, what was the Top value in fitting the non-linear regression? The concentration-response graph does not look as though a plateau in RPRFa effect has been reached. If for example 100 μ M is a maximal effect and the Top value was ~ 0.4 (I3s/lpk), then this may be underestimating the potency. Alternatively, if the RPRFa effect has not reached saturation at all, the EC50 may be less potent. Related to this, in Line 114, regarding the text "higher than previously published (4.23 μ M) (29)". It might be good to add a qualifier to this sentence such as "higher than previously published (4.23 μ M) under similar conditions" – given the cited data is analysing I5s/lpeak and not I3s. Although with the above saturation point in the data presented here, it's uncertain whether the differences are larger or smaller than described.

2) Line 203-208: It reads a little strange to perform the control and +RPRFa activation curve analyses with I/lmax, but then to compare these shifts with the +WRPRFa where the I3s/lpk. From the example traces in Fig 3B, it doesn't look as though the +WRPRFa I/lmax or I3s/lpk data would look that different. I understand the consistency of analyses point in the current, but it

feels as if the comparisons for pH-sensitivity are made between sample and control at different points.

Similar point with Line 406-407 and the SSD data. Axis label for +WRPRFa is also missing for Fig 6E.

Related to this, regarding the concatemer channels for Fig 5D and E, this data is for I/Imax, but is compared against the WT ASIC3 which is the I3s/Ipk data? Again, although the conclusions made likely do not change, it doesn't seem that like-for-like comparisons are being made.

3) Line 269-272 (E78Q data in Fig 4C and D): The wording of the text could be modified a little to improve clarity. It should be made more clear that WRPRFa has a very clear and significant effect on the peak current amplitude of E78Q, this doesn't resemble WT channels at all. The interpretation of "which suggested the mutation had altered channel gating but not diminished its interaction with WRPRFa" also needs some adjusting in writing. From this data alone it's not clear whether the pH 6.3 activating solution used here for the mutants is at the same point of the peptide-free pH-activation curve for mutants vs WT. It could be that the difference in peak current effect could simply be due to the different points pH 6.3 lies on the activation curve. {Reading further can see this is investigated, but maybe also reiterates that it's hard to make the interpretation stated without this knowledge at the earlier point in the manuscript}. Alternatively, the WRPRFa interaction with E78Q could be equal to WT and not diminished at all, and it simply be that the peptide has a different pharmacological outcome at E78Q than WT. I think the interpretation here either has to be caveated with alternative reasons, or simply left out for now and the results presented without interpretation (and this left to later). The interpretation here is also somewhat inconsistent with the stronger wording later in the paragraph stating that "our experiments demonstrate the lower glutamate pair (E78/E421) is important for WRPRFa activity."

Minor comments:

- Lines 43-44: ASIC2b homomers are also not thought to open in response to extracellular acidification, and may be worth noting like has been done for ASIC5 here.
- Line 50: The "wrist" is more typically thought of as the region linking the transmembrane domain and ECD, not so much that the transmembrane domains are the wrist as described in text.
- Line 72-78: When introducing the ASIC3 targeting molecules, both APETx2 and RPRFa are described. Adding the earlier FMRFa paper (PMID 10798398) and the more recent conoRFamides (PMID 31028742) seems to fit here. Particularly as both papers describe selectivity within ASIC subtypes.
- Fig S1: The figure legend has "I3s/Ipk" but the graph y-axis is "I10s/Ipk". If the figure legend is incorrect, to correct the typo.
- Supp Fig legends: All the Supp Fig legends have "x/±" where it's a typo or conversion error for "±"? This also appears in the main text at points (e.g. Line 213 and 217).
- Line 114: "higher than previously published (4.23 μM) (29)" it might be good to add a qualifier to this sentence such as "higher than previously published (4.23 μM) under similar conditions" – given the cited data is analysing I5s/Ipeak and not I3s.
- Line 1052 (Fig S2 legend): could be worth noting in the legend that the QPatch II was used for this comparison. Just to make clear it's not local perfusion as would be possible with the manual rig setup.
- Fig 1G-I: Please confirm/clarify that this data for WRPRFa is using the same setup as in Fig 1A-C, and not the QPatch II.
- Fig S6: The axis labels are missing.
- Line 277-278: Regarding the lack of measurable currents from the glutamate to aspartate mutants, I think it would help to either note in text or as a figure the pH conditions used for these experiments. It could just be that the pH-activation threshold is shifted for the aspartate mutants.
- Line 290-299: If space permits, it could be helpful to the reader to include the typical square WT/WT, WT/MT, MT/WT, MT/MT quadrant box for two double mutant cycle analyses pairs. Potentially easier to read/follow than the text alone and having to draw the entire mental map.
- Line 326-328: Regarding the concatemer channels. Agree that most previous studies have used co-injection or co-transfection, however, similar concatemeric channel designs have been used before. PMID: 27277303, 37422581, or 35156612 for similar ligand binding/stoichiometry studies.

Version 1:

Reviewer comments:

Reviewer #1

(Remarks to the Author)

I appreciate that the authors have substantially improved their manuscript. I have the following remaining comments:

1) The last part of the title ("..., which unveils gating mechanisms.") is no longer appropriate, and the authors should consider changing it.

2) Lines 397-399: "...these results suggest a single ASIC3 subunit contains a sufficient WRPRFa binding site...". I do not agree with this conclusion because it suggests three independent binding sites. It would be safer to say "...these results suggest all three ASIC3 subunits contribute to a WRPRFa binding site...". The same for lines 565-566

3) Figures 7C-E: I do not agree with the interpretation of these results. In Fig. 7C, peak ii is clearly different from peak i, indicating that WRPRFa unbound in between. The ratio I3s/Ipk is an insufficient parameter, as both Ipk and I3s decreased between peaks i and ii. If one would fit the decay of peak i, the decay of peak ii would nicely follow this fit, suggesting that WRPRFa unbound during decay.

I also note that the very slow washout of WRPRFa from the open state suggests that the peptide is trapped in its binding site. Otherwise, if binding/unbinding were simply limited by diffusion, such a slow wash-out would suggest a much higher affinity. Binding site trapping would be compatible with binding to a cavity in the lower palm domain.

4) Lines 552-553: the sentence starting with "Alternatively, ..." is confusing. Please rephrase.

Minor: Line 77: change "As2a" to "ASIC2a"

Reviewer #2

(Remarks to the Author)

The authors have sufficiently addressed most of my concerns. There are a few remaining points, that I list here by using the previous numbering of the comments.

Major points

2.5. Please provide more specific y-axis titles also for Figs. 4F, 4G, 8A and 8C.

3.2. There is a discussion of possibly changed pKa values of residues E78 and E421 (lines 574-586). A change in pKa should also affect the pH dependence. For this reason it is important to discuss the evidence for shifts in pH dependence of mutations at these positions (PMID 17389250 and PMID 22948146).

5. Lines 354-355, "Given the E78/E421 pair is located away from subunit interfaces...", seems wrong to me, since these residues are oriented towards the central cavity, where the three subunits approach each other.

6. Lines 614-621, when discussing previous studies measuring tachyphylaxis, mention also that in the present study, a higher stimulation frequency was used than in previous studies.

New analysis in Fig. 7C-D, lines 419-445. The description of the experiment is difficult to follow; I had to look it up in Reiners et al. to completely understand. Could you reformulate it in a way that it is understandable without going back to the original description of the protocol? In the text lines 443-445, when making conclusions about changes in the peak amplitude in this protocol, provide data. In the legend to Fig. 7C, indicate the timing, since this is not so clear from the figure itself.

Small changes to make (new numbering)

1. Line 77, would "As2a" mean "ASIC2a?"

2. Lines 390-391, "The shift in pH-sensitivity for ASIC313 was less than WT ASIC3 (0.73 units), and ASIC131 was shifted even less than ASIC313, consistent with the number of ASIC3 subunits in the channel." Provide evidence that these values are statistically different from each other.

3. Lines 393-397, similarly to the passage above, provide results of statistical tests to support these statements.

4. Mutant cycle analysis, lines 321-351, WIPRFa is used for testing position R4, while WRP{ADMA}Fa is used for R2. For the analysis of the position E378/E416, WIPRFa is used. Therefore, on Line 347, "the site 2 Arg with.." should rather be "the site 4 Arg with..".

Reviewer #3

(Remarks to the Author)

Thank you to the authors for considering all the reviewer comments. The updated manuscript looks good. Hopefully future work will resolve the still contentious points that have been uncovered by the data in this manuscript.

Version 2:

Reviewer comments:

Reviewer #2

(Remarks to the Author)

The authors have now sufficiently addressed the points that have remained unresolved after the first revision.

We thank all the Reviewers for their help to improve our manuscript. Some changes were extensive and many were prompted by comments from more than one reviewer. Accordingly, we summarize here the major changes to the manuscript for the convenience (and interest) of all Reviewers. Point by point responses are included in the following section with updated line numbers.

-We reexamined the strength of data in support of our claim for stoichiometry of more than 1 peptide per channel and concluded we had insufficient evidence for that claim. We removed statements that suggested our results demonstrated a stoichiometric ratio $> 1:1$.

-The hypothetical model to explain the opposing effects of E78Q and E421Q mutations on WRPRFa potency was removed in favor of more general language regarding glutamate pairs and trinary interactions.

-We removed the comparison of linear and circular gating models as well as their implications on the nature of steady-state desensitization. Although we still believe WRPRFa could be a tool to distinguish the two, a proper treatment of SSD is beyond the scope of this manuscript and the data presented were too limited to draw firm conclusions.

-Several comments made reference to unbinding of WRPRFa from ASIC3 prior to desensitization, which we had not tested in the original manuscript. A previous study by Reiners et al. (2018) demonstrated the RPRFa peptide first needed to unbind from ASIC3 before the channel could desensitize. Thus, the slow component of desensitization originated from peptide dissociation while the fast component was desensitization of the unbound channel. We include the original panel below for comparison. Desensitization of the channel during an acidic stimulus resulted in a decreased fraction of sustained current consistent with dissociation of peptide and subsequent desensitization and recovery of unbound channels. We repeated a very similar experiment with RPRFa and observed the same result. However, with WRPRFa, we observed no significant decrease of the sustained current fraction, suggesting the decrease in current during an acidic stimulus may arise from desensitization of WRPRFa-bound channels. The interpretation we put forward in the text is that the pH-dependence of WRPRFa-treated ASIC3 is a consequence of proton titration of the E78/E421 pair, whose neutralization under acidic conditions weakens the binding between WRPRFa and ASIC3. Unlike RPRFa, which unbinds from the channel, WRPRFa remains associated via its N-terminal Trp and furthermore stabilizes the desensitized state. In lieu of contrasting this result in linear and circular models, we instead use the model established by Reiners et al.

Reproduced from Reiners et al. (2018)

RPRFa on ASIC3, repeated by the authors

WRPRFa on ASIC3

-We revised the text to make clear that our experiments testing the effect of glutamate mutations on WRPRFa in the lower palm of ASIC3 did not define a novel binding site but confirmed overlap with sites already described for RPRFa and FRRFa on ASIC3 and ASIC1a, respectively. However, specific sidechain-sidechain interactions predicted by molecular docking had not been previously validated, which we did here, the results of which are presented in the new Figure 5. Accordingly, we performed additional experiments examining the interaction of the site 4 Arg with the upper glutamates E378 and E416.

-Figure 4 was split into two figures: the new Figure 4 includes the effects of E78Q on pH-dependence of activation while Figure 5 describes the double mutant cycle experiments.

-Table 2 was added to present the parameters of concentration-response curves for different combinations of RFamide and ASIC3 WT or mutant. These include EC_{50} , E_{max} , and Hill slope. This has improved the readability of the manuscript, particularly for the portions of text associated with Figures 4 and 5.

-We inaccurately reported EC_{50} values for concentration-response curves that did not reach or approach a plateau. Now, we constrained the top or E_{max} to the geometric mean of the effect at the highest concentration. This probably overestimates the potency of the peptide on the channel, and we report this value as a lower limit for the potency. This also has the effect of setting an upper limit to the calculated coupling co-efficient.

-HEK cells express endogenous ASIC1 and if expression levels were sufficiently high, these channels could possibly form heteromers with the stably expressed ASIC3. A qRT-PCR experiment demonstrated the transcript level of ASIC3 far exceeded those of ASIC1 (307-fold), such that ASIC3 homomers would constitute 99% of the ASIC population. The ASIC1 probe targeted the exon 10-11 junction and so detects both ASIC1a and ASIC1b transcripts.

Reviewers' comments:

Reviewer #1 (Remarks to the Author):

In this manuscript, Chien et al. report the discovery of a new RFamide peptide, WRPRFa, that modulates acid-sensing ion channel 3 (ASIC3) more strongly than previously described RFamides. They go on to replace systematically individual WRPRFa residues by several other natural and unnatural amino acids to determine the structure-activity relationship. Moreover, they performed experiments to determine the state-dependence of WRPRFa binding to ASIC3. In addition, they replaced individual amino acids on ASIC3 to identify a possible binding site and used double mutant cycle analysis to confirm the interaction of an Arg residue of WRPRFa with an acidic residue of ASIC3. Then they used concatameric ASIC1a-ASIC3 heteromers with a fixed number of ASIC3 and ASIC1a subunits to analyze the influence of the number of ASIC3 subunits on modulation by

WRPRFa. In addition, they used WRPRFa modulation to shed light on two different schemes that had been used to explain ASIC gating. Finally, they report for the first time tachyphylaxis—a reduced response of the channel with repeated activation—for ASIC3. This is an interesting manuscript that contains a lot of information. There is no doubt that WRPRFa much more strongly delays ASIC3 desensitization than previously described RFamides. However, not all conclusions are supported by the data, some claims are overstated, and not all findings are novel.

We thank Reviewer #1 for their constructive criticism and thoughtful recommendations to strengthen our manuscript.

Main comments:

1) In the abstract, the authors claim that the RPRFa peptide “has an unresolved mechanism and binding site”. However, later in the manuscript, it becomes clear that a putative RPRFa binding site has previously been identified by molecular docking and site-directed mutagenesis. The binding site proposed in the present study involves similar amino acids and, thus, at least overlaps with the previously identified binding pocket. This is also correctly stated later in the abstract (“...and identify an interaction site overlapping with other RFamides”). In addition, a mechanism for slowed desensitization by RPRFa had also previously been proposed. These previous findings are largely confirmed in this study. Claims of novelty should therefore be toned down.

We removed the statement on “unresolved mechanism and binding site” from the abstract.

We revised the relevant text within the Abstract [25-27], Introduction [95-96], Results [277-281, 323-326], and Discussion [571-574, 583-586] to clearly acknowledge prior work outlining a binding site and proposing specific sidechain-sidechain interactions, and to indicate that our double mutant experiments validated some of these predictions. We performed additional experiments examining the interaction of the site 4 Arg with the upper glutamates to provide a more comprehensive assessment of proposed sidechain-sidechain interactions.

Furthermore, prompted by comments from Reviewers #1 and #2, we tested if the “unbinding before desensitization” mechanism previously demonstrated by Reiners et al. for RPRFa held true for WRPRFa. As discussed in the section at the beginning of this document, our results suggest WRPRFa may employ a somewhat different mechanism for the slowed rate of desensitization. [421-447, 601-613]

2) The authors used HEK and COS cells in their study. It was not always clear to me which cells were used for which experiments. This should be more clearly stated in the figure legends. HEK cells contain endogenous ASIC1a. Stably expressing ASIC3 in these cells should give rise to at least a subpopulation of ASIC1a-ASIC3 heteromers, which should contaminate measurements. How was a possible contamination by heteromers assessed?

Likewise, it was not clear to me, in which experiments automated patch clamp was used and in which manual patch clamp. This should also be stated more clearly.

We performed a qRT-PCR experiment to compare the copy numbers of ASIC1a and ASIC3 in the HEK ASIC3 stable cell line. We determined that ASIC3 transcript levels were 307-fold higher than ASIC1a, which indicates the ASIC population in HEK stable line should be 99% ASIC3 homomer. We have included a section in the Methods [703-713] and Results [125-130]

“qRT-PCR expression quantification

HEK cells stably expressing ASIC3 were lifted with Detachin and washed once with DPBS without Mg^{2+} or Ca^{2+} , then centrifuged for 10 minutes at 500 x g. RNA was harvested from the cell pellet with an RNeasy Mini Kit (Qiagen, Germantown, MD). An EXPRESS One-Step Superscript qRT-PCR Kit (ThermoFisher) used to setup the reverse-transcription and amplification reactions. The experiment was run on a StepOnePlus Real-Time PCR system (Applied Biosystems, Singapore) and analyzed with StepOne Software v2.3 (Life Technologies). Standard curves for ASIC1a and ASIC3 were made by 1:10 serial dilutions of human ASIC1a (NM_001095.4) and human ASIC3 (NM_004769.4) clones in a pcDNA3.1-P2A vector backbone. Primer-probe assays for ASIC1 (Hs.PT.56a.1262264; exon 10-11) and ASIC3 (Hs.PT.58.24784572; exon 5-7) were purchased from IDT (San Diego, CA).”

“HEK cells are known to express endogenous ASIC1, which could form heteromers with the stably expressed ASIC3 and alter pharmacology (46, 47). We performed a qRT-PCR experiment to measure transcript copy number and determined the level of ASIC3 transcript was 307-fold greater than ASIC1, indicating heteromers would constitute ~1% of the total channel population (Fig S2C-E). Therefore, the level of endogenous ASIC1 in HEK cells is insufficient to affect the interpretation of results.”

We updated the Methods section to indicate all electrophysiology experiments were performed by MPC as a default, unless specifically indicated to have been done on QPatch II. [749-751, 768-769]

“All electrophysiology experiments were performed by manual patch clamp as a default unless otherwise specified.”

“Experiments performed on QPatch II are specifically indicated in the text.”

3) The authors provide a precise EC50 value for RPRFa (66.4 μ M), although the data points do not saturate and only allow to provide an estimate of apparent RPRFa affinity (Fig. 1B). Either measurements should be provided using more RPRFa concentrations or conclusions regarding differential affinities of RPRFa and WRPRFa should be toned down.

We reanalyzed concentration-response data for curves where I_{3s}/I_{pk} did not approach a plateau or reach 1, instead constraining the Top to the geometric mean of the I_{3s}/I_{pk} at the

highest concentration. We then reported this EC_{50} as the lower limit of its true value. We also updated the Methods to explain how EC_{50} s are reported when the concentration-response curve does not saturate. This applied to RPRFa/WT ASIC3, WRPRFa/E421Q, and WIPRFa/WT ASIC3. [787-790]

“For peptide concentration-response curves, Bottom was set to 0 and Top was constrained to ≤ 1 for peptides that approached plateau phase or where I_{3s}/I_{pk} reached 1. For peptides that did not exhibit a plateau, we constrained Top equal to the geometric mean of the highest concentration and reported the EC_{50} as a lower limit of the true value.”

4) How well was the pH in the measurements controlled, in particular for the measurements using automated patch clamp? The authors should provide more details on this. Was there a constant solution exchange or was solution exchanged only once? If so, how much of the solution was exchanged? These details are relevant to judge the quality of the measurements. It was also not clear whether the pH of all solutions containing peptides had been adjusted after adding the peptides. Please state this explicitly. How was the pH measured? In some measurements the authors indicate very precise pH values, for example in Fig. 6B “4.94”, when the “nominal pH” was 5.0. What does this mean?

The QPatch II uses a vacuum pull-through microfluidic system to exchange the bath/extracellular solution in the recording chamber. The recording chamber volume is $\sim 1.7 \mu\text{L}$ and we dispensed 5 μL of solution when changing the bath, which takes about 100 ms to replace 99% of the solution. [765-769]

“The extracellular solution was changed by dispensing 5 μL of buffer through the $\sim 1.7 \mu\text{L}$ recording chamber, with a solution switch time of 100 ms, that exchanges 99% of the solution.”

For manual patch clamp, we used a bath with a $\sim 300 \mu\text{L}$ volume and two perfusion systems. When pre-treating with peptides at pH 8.0, we exchanged the entire chamber by flowing peptide for at least 1 minute. The bath perfusion system has a flow rate of 8 mL/min. For changes in pH (acidification or alkalinization after acidification) we used a focal perfusion system with a flow rate of 3 mL/min through a quartz millimanifold with an internal diameter of 500 μm . The patched cell is positioned in the outlet of the millimanifold, which ensures the cell is entirely bathed in the applied solution. Solutions were continuously flowed through millimanifold after the first focal application. [725-730]

“Peptides used for pre-treatment were applied at pH 8.0 using a bath perfusion system with a flow rate of $\sim 8 \text{ mL/min}$ and flowing for at least 1 minute. All other solutions were focally applied by a Millimanifold perfusion pencil (ALA Scientific, Farmingdale, NY) with a flow rate of $\sim 3 \text{ mL/min}$. The patch-clamped cell was positioned in the outlet of the perfusion pencil and solution was flowed continuously throughout the experiment to maintain the applied solution and pH around the cell.”

The pH of solutions was measured with an Orion STAR A series pH meter while creating the solution and again before starting experiments. We made solutions to a “nominal” pH (e.g. pH 5.0) but reported the measured pH (e.g. pH 4.94) as pH values can drift over time despite buffering. We observed no effects of peptides on solution pH even up to 100 μ M, making it unnecessary to further adjust pH. [735-737, 742-743]

“Bath solution pH was measured with an Orion STAR A series pH meter (Thermo) before recording and the measured pH reported.”

“Addition of peptides up to and including 100 μ M did not affect the bath solution pH.”

5) Tachyphylaxis. It is surprising that the authors, for the first time, report tachyphylaxis for ASIC3. If I see it correctly, previous studies used pH 5.0 and pH 5.6 to assess tachyphylaxis of ASIC3, not 5.5-6.0 as stated in line 572. Why has tachyphylaxis not been noticed in these previous studies? Different expression systems as a possible source for this variability should be discussed. Is it possible that “tachyphylaxis” in the present study is due to a more classical “rundown”? Another potential issue is solution exchange. Was it complete? Or is it possible that the acidic pH to activate ASIC contaminated the conditioning pH? If so, experiments with a more complete solution exchange should be performed.

We corrected the text to reflect the correct stimulus pH for prior studies that tested for ASIC3 tachyphylaxis [616-617] and suggest distinct expression systems as a source for variability between this study and prior ones [618-623]. We address solution exchange and control of applied pH in the response to point 4, as well as the changes made to the text.

“Previous studies that reported the absence of ASIC3 tachyphylaxis used stimuli of pH 5.0-5.6 (12, 57), consistent with our observation of minimal tachyphylaxis at pH 5.0. One explanation for the absence of tachyphylaxis in prior studies could be insufficiently acidic stimuli; we saw ASIC3 tachyphylaxis only at pH 4.0, suggesting there may be a threshold concentration of protons required for tachyphylaxis to occur. Additionally, both prior studies used *Xenopus* oocytes for expression of ASIC3 in contrast to the HEK cells we used in this study. Differences in membrane composition or intracellular conditions may influence the extent or pH-sensitivity of tachyphylaxis.”

6) Stoichiometry: the authors largely confirm a WRPRFa binding site in the lower palm domain as previously proposed for RPRFa on ASIC3 and other RFamides on ASIC1a. It should be noted, however, that previous studies reported binding of RFa peptides in a single cavity at the center of the threefold symmetry of the channel. Molecular docking confirmed a single RPRFa molecule in this cavity. The results of the authors do not contradict this view. Therefore, either the authors should provide compelling evidence for the binding of three peptides simultaneously to ASIC3 (is there enough space to accommodate three peptides at once?) or they should change their interpretation.

The interaction of E78 with site 2 of WRPRFa has already previously been proposed by docking results. This finding is therefore not completely novel but rather confirmatory. This should be mentioned.

We reevaluated the strength of our conclusions on RFamide/ASIC stoichiometry and removed conclusions supporting a >1:1 stoichiometry. We changed the text of both Results and Discussion sections. [399-401, 561-563]

7) For the interpretation of some measurements, it would be helpful to describe the action of WRPRFa more carefully. For example, it appears that at 30 μM WRPRFa—the concentration used in most experiments (the concentration should be indicated on the figure)—not all channels had bound WRPRFa (Fig. 1I). I think the initial current decline arises from a mix of channels with no WRPRFa bound undergoing desensitization and of unbinding of WRPRFa from other channels. Moreover, it appears that at acidic pH, the peptide unbinds much more quickly (Fig. 3B). Thus, binding of WRPRFa to ASIC3 is incomplete and complex. This caveat should be kept in mind when interpreting experiments designed to shed light on gating mechanisms.

We agree the effects of WRPRFa on ASIC3 can be difficult to interpret. Not all channels are bound and some fraction of current does appear to desensitize very early in the activation period, which could be due to desensitization of unbound (“never bound”) channels or unbinding followed by desensitization. For the first case, we used I_{3s} to measure currents arising from bound channels, as “never bound” channels would have desensitized by this time. In the second case, we repeated an experiment that had been previously performed by Reiners et al. (2018) for RPRFa that demonstrated the peptide needs to first unbind before desensitization. In our study, WRPRFa did not yield the same outcome, suggesting the mechanism underlying the slow component of desensitization may differ between the two peptides. We believe this to be related to the pH-dependence of desensitization observed for WRPRFa and discuss it in the Results [421-447] and Discussion [601-613] sections, as well as at the beginning of this document.

8) The authors used WRPRFa to investigate gating mechanisms of ASIC3. I could not follow all their arguments and found some interpretations not convincing.

8.1 First, they find that the pH50 of activation strongly shifted to the left so that it became very similar to the pH50 of steady-state desensitization (SSD). It appears to me that this is exactly what a linear gating scheme shown in Fig. 6C would predict for a substance that almost abolishes desensitization.

We agree with this interpretation. However, we have removed the comparison of SSD models and linear vs circular gating schemes, thus the referenced text is no longer part of the manuscript.

8.2 In lines 365-366, they state that “acute desensitization happens under more acidic conditions”. This is incorrect. In the simplest case, acute desensitization happens with a

fixed rate constant irrespective of pH. It may appear slower at higher pH because not all channels desensitize at the same time.

We have adjusted the text to indicate that acute desensitization is more apparent under acidic conditions. [404-407]

“Acute desensitization is pH-independent and terminates channel activation while SSD occurs at sub-activating pH levels without apparent channel activation. Additionally, SSD shows pH-dependence, where desensitization becomes more complete as the stimulus grows more acidic (1).”

8.3 In lines 390-391, they state that SSD of WRPRFa-bound channels should not be possible under the linear model. It was not clear to me why it should be possible in a circular model. It appears that WRPRFa (largely) prevents desensitization. If there is a single “D” state in both models, why can it be reached by WRPRFa-bound channels in the circular model but not the linear model?

We have revised the section on desensitization to remove comparison of SSD models, linear vs circular schemes to instead focus on the differences between RPRFa and WRPRFa unbinding using an established gating model.

8.4 The authors attributed the SSD to the difference in I3s between the conditioned test pulse and baseline (lines 416-418). How can they exclude that this difference is in part due to unbinding of the peptide at the more acidic pH? The interpretation of the whole set of these experiments is confusing and not convincing. In my opinion, SSD should be estimated from the decline of the peak current, because it arises (at least in part) from acute desensitization of channels that had no peptide bound. If the peptide prevents desensitization, there should be no SSD in both models. I doubt that this peptide can be used to convincingly differentiate between a linear and a circular model.

We have revised the section on desensitization to remove comparison of SSD models, linear vs circular schemes to instead focus on the differences between RPRFa and WRPRFa unbinding using an established gating model.

9) The experimental design for some experiments shown in Figure 3 is not optimal. In Fig. 3D, there is no time interval between peptide application in the conditioning period and during activation at pH 6.3 Thus, the measurement on the right of the trace is not “the same” (line 226) and not directly comparable to the one on the left of the same trace. In Fig. 3F and G, to determine the off-rate from resting channels, it would have been better to use different times of wash-out followed by a single activation (and not to regularly activate ASIC3, even not for 5 sec). Perhaps it is better to show the experiment to determine the off-rate from activated channels first, because it allows to estimate the error introduced by repetitively activating channels.

Figure 3D (now Figure 3G) compares the binding that occurred during the first stimulus (application of WRPRFa to the desensitized state, measured by I_{3s}/I_{pk} in the second stimulus) with the amount of binding that occurs to the resting state (application of WRPRFa before the third stimulus, measured by I_{3s}/I_{pk} in the third stimulus) to demonstrate the extent of binding to the desensitized state is much less than that of the resting state. These are not “the same” in the sense that they are identical applications and stimuli, but we intended they are the “same” experiment in that they are performed on the same cell as part of a single protocol.

In the above experiment, there was a brief recovery interval between stimuli 1 and 2 to allow channels to recover from desensitization during which peptide was not applied lest it bind to the recently recovered, resting channels. An interval of the same duration without WRPRFa between pre-application and stimulus 3 was not present, but the results from (the new) Figure 3D make it apparent that very little WRPRFa would have washed off from the bound, resting state channels.

We rearranged Figure 3 to present the kinetic experiments first, followed by the state-dependence experiments. We present the activated state wash-off before the resting state wash-off to demonstrate the negligible wash-off from the activated state during the 5 second pH 6.3 applications.

10) The paragraph starting with line 548 is highly speculative and should be deleted or strongly shortened. If this model is correct, I would expect that WRPRFa unbinds during activation because E421 is no longer charged.

We removed both the figure and the accompanying Discussion section.

Minor comments:

1) Line 50: “with the first transmembrane domains as the wrist”. I think it should be the forearm. The wrist is between the transmembrane domains and the ECD.

We emended this sentence to correctly identify the wrist. [47-49]

“Between the two transmembrane segments is a large extracellular domain (ECD) whose structure has been described as a “closed-fist” with the transmembrane domains connected to the ECD by a “wrist”.”

2) At several instances, more references would aid the reader not so familiar with the literature. For example, line 86 (references for the binding site in the lower palm domain),

We have added references for prior studies identifying a binding site in the lower palm domain, among others. [85-87]

“Structural studies reveal FaNaCs bind their ligand FMRFa in the distal finger domain but docking and mutagenesis studies indicate FRRFa and RPRFa bind the lower palm domain in ASIC1a (43, 44) and ASIC3 (45), respectively”

3) Line 94: “revealed distinctions”. Please be more specific.

We rewrote this sentence to align with our changes per main comment #1. [96-96]

“Mutagenesis of WRPRFa and the ASIC3 lower palm domain confirmed a specific intermolecular contact between the peptide and channel”

4) Lines 161 and 188: the authors underline the “ringed side chain” of a His residue. I think given that the imidazole group can be protonated, it is not so curious that His behaved different than other aromatic amino acids.

We edited both sentences to restrict comment to the imidazole side chain. [174-176, 200-201]

5) Lines 264-266: “we saw four glutamates in two pairs”. The pairs cannot be clearly seen in Fig. 4B. Only later (in lines 291-292), the authors mention previous crystal structures that revealed these pairs. These studies should be mentioned and cited earlier to make the notion of acidic pairs more convincing.

We added an earlier mention of the studies supporting the presence of glutamate pairs. [289-291]

“The lower two glutamates (E78/E421) have been previously described to form a pair, with their side chains in close approximation (6, 48).”

6) Lines 326-328: There are now several published studies that used ASIC concatamers. Therefore, I recommend deleting this sentence, which is somewhat diminutive with respect to previous work by others. Simply state why you used concatamers.

We included references highlighting prior work using concatamers to enforce ASIC subunit stoichiometry (helpfully provided by Reviewer #3). We restrict our statement on prior studies using co-injection only to those studying interaction of RFamides with ASIC heteromers. This is relevant as prior studies could not characterize the effect of RFamides on specific heteromers (e.g. ASIC131 and ASIC313) as we have done here. [367-372]

“Prior studies of RFamide interaction with ASIC heteromers have used co-injection into oocytes, which does not allow isolation of a single heteromeric species (30, 52–54). To overcome this limitation, we generated heteromeric concatemer channels of ASIC3 and ASIC1a subunits, using the latter as insensitive replacements for binding-competent ASIC3 subunits. This is an approach that has been used successfully in the past to generate

diverse heteromeric ASICs (5, 55, 56).”

7) Line 338: Delete “accelerated”

We removed “accelerated” from this sentence. [379-380]

Reviewer #2 (Remarks to the Author):

This study investigates the function of a short peptide that selectively modulates ASIC3. The authors try different modifications of the known ASIC3 modulator RPRFa, and find that N-terminal addition of a Trp residue strongly increases the effect of the peptide, leading at 100 uM to a completely non-desensitizing current. They test how exchanging residues at either of the five positions of WRPRFa affects modulatory properties of the peptide, identifying different, position-dependent sensitivities to such changes. Of the potential interacting residues in the binding site on ASIC3, mutations of acidic residues were tested, identifying E421 as important for the modulatory effect of the peptide. Experiments with concatemeric constructs indicate that one WRPRFa-binding subunit in the trimer is sufficient for the peptide to exert full modulation of ASIC3 desensitization. Finally, the authors observe tachyphylaxis that is increased by WRPRFa, if they stimulate ASIC3 at high frequency with very acidic pH solutions. Overall, this is an interesting description of a new ASIC3-modifying peptide.

We thank Reviewer #2 for their helpful criticism and attention to detail.

Major points

1. The analysis of the ASIC3 pH dependence showed a substantial alkaline shift in the presence of WRPRFa. In the peptide-free condition, the test pH corresponds to the pH50. For this reason, the exposure to the peptide should, based on the pH50 shift, lead to a 2-fold increase in the peak current. This is evident in some of the current traces shown, but not in others. A possible change in peak current upon peptide exposure is not measured quantitatively. Could the authors provide such an analysis in the manuscript and interpret the finding?

We revisited the data from our QPatch II peptide screen and examined the increase in peak current after application of 30 uM WRPRFa. We calculated $I_{pk, WRPRFa} / I_{pk, vehicle}$ as a measure of the fold-increase in peak current caused by the alkaline shift in pH-sensitivity induced by WRPRFa. As predicted, the geometric mean of the fold-increase was 2.26-fold \times/\div 1.19.

We added this result as Figure S8B and discussed it in the text. [221-227]

“As WRPRFa caused an alkaline shift in the pH-sensitivity of ASIC3 activation, treatment with the peptide should increase the current magnitude in response to a fixed acid stimulus. The pH-dependence of activation described above indicates pH 6.3 activated about 55.7% of unbound channels and 99.7% of WRPRFa-bound channels,

which should result in a 2-fold increase in current after WRPRFa binding. We examined the fold-change in peak current after WRPRFa treatment during the experiment depicted in Figure 1F. The measured fold-change was $2.26 \times / \div 1.19$ (Fig. S8B), consistent with our prediction.”

2. Methodological details: The description of some experiments is not clear enough, as listed below. These aspects should be clarified.

2.1. Line 105, Many subtypes of HEK cells contain endogenous ASIC1a. Indicate in the methods whether you can confirm that your cell line does not express ASIC1a.

We performed a qRT-PCR experiment to compare the copy numbers of ASIC1a and ASIC3 in the HEK ASIC3 stable cell line. We determined that ASIC3 transcript levels were 307-fold higher than ASIC1a, which indicates the ASIC population in HEK stable line should be 99% ASIC3 homomer. We have included a section in the Methods [703-713] and Results [125-130]

“qRT-PCR expression quantification

HEK cells stably expressing ASIC3 were lifted with Detachin and washed once with DPBS without Mg^{2+} or Ca^{2+} , then centrifuged for 10 minutes at 500 x g. RNA was harvested from the cell pellet with an RNeasy Mini Kit (Qiagen, Germantown, MD). An EXPRESS One-Step Superscript qRT-PCR Kit (ThermoFisher) used to setup the reverse-transcription and amplification reactions. The experiment was run on a StepOnePlus Real-Time PCR system (Applied Biosystems, Singapore) and analyzed with StepOne Software v2.3 (Life Technologies). Standard curves for ASIC1a and ASIC3 were made by 1:10 serial dilutions of human ASIC1a (NM_001095.4) and human ASIC3 (NM_004769.4) clones in a pcDNA3.1-P2A vector backbone. Primer-probe assays for ASIC1 (Hs.PT.56a.1262264; exon 10-11) and ASIC3 (Hs.PT.58.24784572; exon 5-7) were purchased from IDT (San Diego, CA).”

“HEK cells are known to express endogenous ASIC1, which could form heteromers with the stably expressed ASIC3 and alter pharmacology (46, 47). We performed a qRT-PCR experiment to measure transcript copy number and determined the level of ASIC3 transcript was 307-fold greater than ASIC1, indicating heteromers would constitute ~1% of the total channel population (Fig S2C-E). Therefore, the level of endogenous ASIC1 in HEK cells is insufficient to affect the interpretation of results.”

2.2. Please indicate in the methods, which data were obtained with manual patch-clamp, and which with the Qpatch II.

We updated the Methods section to indicate all electrophysiology experiments were performed by MPC as a default, unless specifically indicated to have been done on QPatch II. [749-751, 768-769]

“All electrophysiology experiments were performed by manual patch clamp as a default unless otherwise specified.”

“Experiments performed on QPatch II are specifically indicated in the text.”

2.3. For the experiments with QPatch II, indicate how the cells were prepared for the measurement, specify also the low-pass filtering applied.

We updated the Method section to describe cell preparation for QPatch II experiments [756-760] and included the acquisition and low-pass filtering frequencies [767-768].

“Prior to recording, the cells were washed twice in PBS (Gibco, Paisley, UK) and treated with Detachin (Genlantis, San Diego) for 3 minutes, then resuspended in their respective growth media without serum for a concentration of 2×10^6 cells/ml. A minimum of 5 mL of cells were added to the QStirrer, which was placed on the deck of the QPatch II and allowed to incubate at room temperature for a minimum of 20 minutes.”

“Signals were acquired at 1 kHz and low-pass filtered at 300 Hz”

2.4. Indicate how concentration-response curves (for peptide and pH) were obtained, whether full curves were obtained on single cells or whether a limited number of data points were measured on each cell. In case of the second variant, indicate how data points from different experiments were combined. In the figure legends, the number of full, independent experiments should be indicated. If concentration-response curves were pooled from different experiments, indicate also how statistical comparison between different conditions was done (for example pH50 with or w/o the peptide).

We assembled concentration-response curves for pH_{50} , EC_{50} , and τ by pooling data from multiple cells and fitting a single curve to the pooled data. Loss of seals and a limited number of solution reservoirs prohibited collection of full curves from single cells, so limited data points from multiple cells (the “second variant”) were combined in a single plot. pH-dependence of activation or desensitization experiments were normalized to a single pH before adding to the plot, while peptide concentration-response data reported only I_{3s}/I_{pk} . We included additional description of this process to the Methods section and the number of cells used for each plot in the figure legends. [812-817]

“Data points from multiple cells were combined into a single plot before curve fitting. For pH-dependence of activation or time-dependence experiments, data from each cell were normalized to an indicated stimulus before addition to the combined plot. For peptide concentration-dependence of activity, the I_{3s}/I_{pk} value was calculated before addition to the combined plot. The number of cells tested for each condition in a plot is indicated in the figure legend.”

Comparison between different conditions was made with Akaike’s Information Content (corrected), which describes the probability that the indicated parameter is different between the two conditions. If of interest, a background on AICc can be found here

(https://cdn.graphpad.com/faq/2/file/Prism_v4_Fitting_Models_to_Biological_Data.pdf) on page 143. [818-822]

“All curves represent global fits to the plotted data, and values derived from fitting approaches are reported in the text with 95% confidence intervals. Parameters for global curve fits (i.e. pH_{50} , EC_{50} , τ) were compared with Akaike’s Information Criterion (corrected) and we reported the probability that the indicated parameter was different between the compared fits (62).”

2.5. On the axis titles and figure legends make it clear, what is really plotted. "I/I8.0" is not sufficient. It needs to be indicated whether I is the peak current or the current after 3s. Provide the timing information, i.e. when exactly an amplitude was measured, in the legends.

We updated Figure 6E (now 7F) to include more descriptive axis labels and added more information to the figure legend. [1152-1158]

3. Logics of mechanistic conclusions.

3.1. State dependence. Lines 209-224, the I3s/Ipk ratio is compared between the protocols in which peptide is applied to the closed, open or desensitized channels (Fig. 3). Although this ratio is higher after exposure to the desensitized than to the open channel, the interpretation given suggest a more important binding to open than to desensitized channels. In comparison, the binding to the closed channel would be more important. Fig. 3F shows then that the peptide unbinds rapidly from closed channels, but not from open channels. Could the authors provide a conclusion that takes all these findings into account? The fact that association is quite slow plays probably an important role in the outcome of the experiments of Fig. 3.

We restructured Figure 3 and the accompanying Results text (per specific comment 7). We edited the text to ensure there is no suggestion that binding to the activated state was more important than binding to the desensitized state. We added a paragraph to the Discussion section to reconcile the apparently slow activated state on-rate with the much slower activated state off-rate. [547-558]

“Experiments to assess state-dependent binding indicate that WRPRFa preferentially interacts with the resting/closed state over the activated and desensitized states, yet WRPRFa dissociated from activated channels much more slowly than resting state channels. While binding to the activated state appeared negligible, the window of time for WRPRFa to bind the activated state before desensitization may be too short if the activated state on-rate is like that of the closed state. ASICs undergo structural rearrangement during desensitization (44, 57) that may conceal or reconfigure the RFamide binding site, preventing binding in both the activated state and desensitized state protocols. Alternatively, the activated channel could adopt a structure that limits access of the RFamide binding site to the extracellular region. This would prevent WRPRFa from

reaching its binding site as well as slowing its dissociation from the channel. High resolution structures of ASIC3 bound to WRPRFa under neutral and acidic conditions would be useful to interpret these results.”

3.2. The hypothesis presented on lines 548-563 is not consistent with measurements in several published studies, that mutations of E78 or E421 result in only small shifts in pH dependence. It should be revised. It was however observed in ASIC3, that the mutation E79A (corresponding to E78 in the present study) induces a strong alkaline shift in the pH dependence of SSD (PMID: 17389250). This might be a promising basis on a hypothesis on the mechanism by which this domain is involved in SSD.

We have removed this figure as well as the accompanying section of the Discussion.

4. Measurement of the pH dependence of activation and SSD. In some experiments (Fig. 3A, lines 204-208), the activation pH dependence on the peptide-exposed channels is measured not as the pH dependence of the peak current, but the pH dependence of the I_{3s}/I_{pk} ratio. In other experiments, as for the concatemers, the analysis seems to have been done on the peak current, but this is not completely clear. While I understand the reasoning behind it, this is not correct. It is ok to determine the pH dependence of the I_{3s}/I_{pk} ratio as an additional measurement, but this does not correspond to the pH dependence of the channel activation of channels exposed to the peptide. For this, the pH dependence of the peak value has to be reported. A similar analysis was done for the SSD, as described in the text (lines 400-407), although this is not completely clear in the corresponding Fig. 6E.

We are grateful for the help in catching this error. In Figure 3A, 6D, 6E, and 7F we plot the $I_{3s}/I_{3s,max}$ for the peptide-treated channels, but the text incorrectly described the peptide-treated measure as I_{3s}/I_{pk} . We corrected this mistake in the text and made the distinction clearer for Figures 6D, 6E, and 7F in the text, figure legends, and figures.

5. Discussion of possible binding sites and number of peptides that bind to one channel trimer. Line 316 ff, it is stated that E78 and E421 are located away from subunit interfaces. However, these residues point towards the central cavity of the channel, and there is a clear possibility that a peptide may interact with more than one subunit. The comparison with binding of such peptides to FaNaC and HyNaC (lines 318-321) is misleading if it is not indicated that in these peptide-gated channels, the binding site is in a completely different channel domain. The experiments with concatemers show that maximal peptide effects in channels containing 1 or 2 ASIC3 subunits are as high as in ASIC3 trimers, but that the concentration dependence is shifted. Some of these points are taken up in the discussion (lines 508-518) and should be adapted. Towards an understanding of the number of peptides binding, could the authors check whether the structures could accommodate three WRPRFa peptides in a single trimer without clashes? This information would be

useful for the interpretation.

We modified the comparison to FaNaC and HyNaC binding modes to clarify they are at a site distinct from that of ASICs. [361-362, 560-561]

We reevaluated the strength of our conclusions on RFamide/ASIC stoichiometry and removed conclusions supporting a >1:1 stoichiometry. We changed the text of both Results and Discussion sections. [399-401, 561-563]

6. Tachyphylaxis experiments. Tachyphylaxis was observed when the channels were activated with high frequency, every 20s. Tachyphylaxis consists in an irreversible loss of function, and it is stated in the text (line 438 and elsewhere) that the loss was irrecoverable. The corresponding data are however not shown. Is this really tachyphylaxis, or just accumulation of channels in the desensitized state? The authors have to test whether with longer recovery periods at pH8, some of the current can be recovered. In this context, the time course of recovery from desensitization is very important. This is shown in ASIC3 in the absence of peptide in Fig. S10. To understand the possible mechanism by which WRPRFa contributes to tachyphylaxis, it is essential to provide also data on the recovery in the presence of the peptide.

We have included a new Figure S10 showing that the current lost to repetitive stimulations at pH 4.0 cannot be recovered during an interval as long as 1 minute. We address the possibility that slowly recovering WRPRFa-bound desensitized channels could underlie the accelerated tachyphylaxis in the presence of WRPRFa. [633-637]

“Alternatively, the results presented in Figure 7C and D suggest the WRPRFa-bound desensitized ASIC3 may have a slowed recovery from desensitization; thus, the accelerated tachyphylaxis in the presence of WRPRFa could be a combination both of tachyphylaxis and accumulation of WRPRFa-bound channels in the desensitized-bound state.”

We address the recovery of desensitization of bound channels in the Results [421-447] as well as at the beginning of this document.

Specific comments

1. Line 43, BASIC, sometimes called ASIC5, does not belong to the ASIC subfamily.

We removed mention of ASIC5. [41-42]

“The ASIC family is composed of four genes, ASIC1-4, that give rise to six distinct isoforms: ASIC1a, 1b, 2a, 2b, 3, and 4.”

2. Lines 43ff, "All isoform...", Homotrimers of ASIC2b or ASIC4 do not respond to extracellular acidification.

We updated the sentence and added references to support the acid insensitivity of ASIC2b and ASIC4. [42-43]

“All isoforms respond to extracellular acidification except homomeric ASIC2b (3) and ASIC4 (4).”

3. Line 58 "neutral pH", I guess you mean pH7.4. "Physiological pH" might therefore be more appropriate.

We changed “neutral” to “physiological”. [56-57]

4. For several experiments, EC50 values are indicated, where no saturation of the pH dependence is observed. This may be ok if the I_{3s}/I_{pk} hits the maximum of 1, but for other experiments it is not justified, as in Fig. 1A, and some of the data in Figs. 4F-I, an EC50 value cannot be determined and can be at best indicated as lower limit of the EC50.

We reanalyzed concentration-response data for curves where I_{3s}/I_{pk} did not approach a plateau or reach 1, instead constraining the Top to the geometric mean of the I_{3s}/I_{pk} at the highest concentration. We then reported this EC₅₀ as the lower limit of its true value. We also updated the Methods to explain how EC₅₀s are reported when the concentration-response curve does not saturate. This applied to RPRFa/WT ASIC3, WRPRFa/E421Q, and WIPRFa/WT ASIC3. [787-790]

“For peptide concentration-response curves, Bottom was set to 0 and Top was constrained to ≤ 1 for peptides that approached plateau phase or where I_{3s}/I_{pk} reached 1. For peptides that did not exhibit a plateau, we constrained Top equal to the geometric mean of the highest concentration and reported the EC₅₀ as a lower limit of the true value.”

5. Line 113, the difference to the previous study might be due to ASIC3 from different species, rat vs. human.

We included a sentence offering explanations for the different EC₅₀ values between our study and the prior study. [117-120]

“The discrepancy in EC₅₀ could arise from species or methodological differences: our study measured the fraction of sustained current at 3 seconds post-peak using human ASIC3 while the prior study used rat ASIC3 and measured sustained current at 5 seconds.”

6. line 167, "Ile and Asn were...", clarify this sentence and provide statistical support for the conclusion.

We removed this sentence.

7. The kinetics of the peptide binding and unbinding (Fig. 3E-F) are important observations, and in my view, they are also required to better understand the state dependency protocols of Fig. 3C-D. It would seem to me more logical to place them before panels C and D.

We rearranged Figure 3 and the accompanying Results section to position kinetic experiments before state-dependence experiments.

8. line 310, what does "46.5% pH50 are different" mean?

This describes the probability that the indicated parameter (pH_{50} in this case) is different between the two conditions by comparing them with Akaike's Information Criterion, as described in the Methods section. [818-822]

9. The part about the two kinetic models (lines 408-423) is hard to follow. Consider rewriting it.

We have removed the part describing the different kinetic models as part of our revision to the section on desensitization. Instead we examine our findings through the lens of an establish gating model. [438-447]

10. Line 432, the cited study investigated ASIC1a, not ASIC3

The cited study primarily studied ASIC1a but Fig S1 of that study presents the effect of a mutation on rASIC3 tachyphylaxis, which is to what we refer.

11. Overall, the discussion on lines 571-595 is very long and could be shortened.

We shortened this section of the Discussion, instead focusing on alternative explanations for our results, or reconciling our findings with those of prior works. [616-640]

"We show here that ASIC3 can undergo tachyphylaxis at pH 4.0. Previous studies that reported the absence of ASIC3 tachyphylaxis used stimuli of pH 5.0-5.6 (12, 57), consistent with our observation of minimal tachyphylaxis at pH 5.0. One explanation for the absence of tachyphylaxis in prior studies could be insufficiently acidic stimuli; we saw ASIC3 tachyphylaxis only at pH 4.0, suggesting there may be a threshold concentration of protons required for tachyphylaxis to occur. Additionally, both prior studies used *Xenopus* oocytes for expression of ASIC3 in contrast to the HEK cells we used in this study. Differences in membrane composition or intracellular conditions may influence the extent or pH-sensitivity of tachyphylaxis.

The rate at which ASIC1a undergoes tachyphylaxis is dependent on the stimulus pH as well as the time spent in the open state, but not the desensitized state. Contrarily, ASIC3 displays a similar loss of current from both the activated/open (Fig. 8A) and desensitized states (Fig. 8C). In both cases the rate of tachyphylaxis in ASIC3 is pH-dependent and

directly related to time at an acidic pH. This discrepancy raises the question of whether ASIC1a and ASIC3 are undergoing the same or distinct processes.

How does WRPRFa accelerate the loss of current under both pulsed and static protocols? One explanation is that slowed desensitization by WRPRFa prolongs time in the open state at a low pH under both protocols, increasing the time for the process underlying tachyphylaxis to occur. In this way, WRPRFa promotes an ASIC1a-type tachyphylaxis. Alternatively, the results presented in Figure 7C and D suggest the WRPRFa-bound desensitized ASIC3 may have a slowed recovery from desensitization; thus, the accelerated tachyphylaxis in the presence of WRPRFa could be a combination both of tachyphylaxis and accumulation of WRPRFa-bound channels in the desensitized-bound state. Finally, tachyphylaxis refers to a decrease in activity after repeated doses or stimuli, which is inconsistent with our observations that ASIC3 currents can be lost irrevocably during a single activation. Future studies exploring these mechanisms may yield more appropriate terminology.”

12. In many figure legends, the concentration-response experiments are called "EC50 for.". Similarly, it is said "dotted lines depicts the RPRFa EC50.". This is not the EC50, but rather the concentration-response curve.

We updated the figure legends to change “EC₅₀” to “concentration-response curve”. [1038, 1047, 1106, 1107, 1109, 1111, 1114, 1134, 1242]

13. Fig. 1A, I am surprised about the kinetics and the amplitude of the ASIC2a current at pH6.3. Can the authors double-check that this is indeed ASIC2a, and that the pH was 6.3?

We tested ASIC2a with a pH 4.0 stimulus but this was not reflected in the figure legend. We updated the Figure 1, S1, and S4 legends to identify the appropriate pH used for ASIC2a stimulation. [1036, 1047, 1202, 1233]

14. Fig. 6F, is "current decrease" relative to the preceding pH or relative to pH8?

Originally this referred to the current decrease relative to pH 8. However, we have removed this figure panel as well as the accompanying section of the Results as we revised our section on desensitization.

15. Fig. 7C, what are the solid lines?

The solid lines were meant to depict stronger interaction between the two sidechains. However, we have removed this figure as well as the accompanying section of the Discussion.

16. Overall, it would be good to increase the line thickness of the symbols in the figures, as in the present version, it is not easy to recognize the different colors.

We have increased the thickness of all lines from 0.25 to 0.5 points.

Reviewer #3 (Remarks to the Author):

This manuscript by Chien and colleagues describes the pharmacological and biophysical characterisation of a synthetic peptide, WRPRFa, that alters desensitisation properties of ASIC3-containing channels. ASICs remain an active area of investigation given their links to sensory physiology, pain pathways, and other diseases involving acidosis. Developing ligands that selectively modify specific gating properties is valuable not only for dissecting ASIC function and mechanisms, but also for distinguishing contributions of different subunit assemblies. In this context, the study positions WRPRFa as a potentially useful pharmacological probe with implications for both basic research and future drug discovery.

Overall, this is a nice paper that would make a good contribution to the literature. Below are a few comments for the authors to consider, along with some minor points listed.

We thank Reviewer #3 for their generous comments and literature recommendations.

1) Fig 1A-C (Line 113): With the reported EC₅₀ of 66 μ M, what was the Top value in fitting the non-linear regression? The concentration-response graph does not look as though a plateau in RPRFa effect has been reached. If for example 100 μ M is a maximal effect and the Top value was \sim 0.4 (I_{3s}/I_{pk}), then this may be underestimating the potency. Alternatively, if the RPRFa effect has not reached saturation at all, the EC₅₀ may be less potent.

Related to this, in Line 114, regarding the text “higher than previously published (4.23 μ M) (29)”. It might be good to add a qualifier to this sentence such as “higher than previously published (4.23 μ M) under similar conditions” – given the cited data is analysing I_{5s}/I_{peak} and not I_{3s}. Although with the above saturation point in the data presented here, it’s uncertain whether the differences are larger or smaller than described.

We reanalyzed concentration-response data for curves where I_{3s}/I_{pk} did not approach a plateau or reach 1, instead constraining the Top to the geometric mean of the I_{3s}/I_{pk} at the highest concentration. We then reported this EC₅₀ as the lower limit of its true value. We also updated the Methods to explain how EC₅₀s are reported when the concentration-response curve does not saturate. This applied to RPRFa/WT ASIC3, WRPRFa/E421Q, and WIPRFa/WT ASIC3. [787-790]

“For peptide concentration-response curves, Bottom was set to 0 and Top was constrained to \leq 1 for peptides that approached plateau phase or where I_{3s}/I_{pk} reached 1. For peptides that did not exhibit a plateau, we constrained Top equal to the geometric mean of the highest concentration and reported the EC₅₀ as a lower limit of the true value.”

2) Line 203-208: It reads a little strange to perform the control and +RPRFa activation curve analyses with I/I_{max} , but then to compare these shifts with the +WRPRFa where the I_{3s}/I_{pk} . From the example traces in Fig 3B, it doesn't look as though the +WRPRFa I/I_{max} or I_{3s}/I_{pk} data would look that different. I understand the consistency of analyses point in the current, but it feels as if the comparisons for pH-sensitivity are made between sample and control at different points.

Similar point with Line 406-407 and the SSD data. Axis label for +WRPRFa is also missing for Fig 6E.

Related to this, regarding the concatemer channels for Fig 5D and E, this data is for I/I_{max} , but is compared against the WT ASIC3 which is the I_{3s}/I_{pk} data? Again, although the conclusions made likely do not change, it doesn't seem that like-for-like comparisons are being made.

We are grateful for the help in catching this error. In Figure 3A, 6D, 6E, and 7F we plot the $I_{3s}/I_{3s,max}$ for the peptide-treated channels, but the text incorrectly described the peptide-treated measure as I_{3s}/I_{pk} . We corrected this mistake in the text and made the distinction clearer for Figures 6D, 6E, and 7F in the text, figure legends, and figures.

3) Line 269-272 (E78Q data in Fig 4C and D): The wording of the text could be modified a little to improve clarity. It should be made more clear that WRPRFa has a very clear and significant effect on the peak current amplitude of E78Q, this doesn't resemble WT channels at all. The interpretation of "which suggested the mutation had altered channel gating but not diminished its interaction with WRPRFa" also needs some adjusting in writing. From this data alone it's not clear whether the pH 6.3 activating solution used here for the mutants is at the same point of the peptide-free pH-activation curve for mutants vs WT. It could be that the difference in peak current effect could simply be due to the different points pH 6.3 lies on the activation curve. {Reading further can see this is investigated, but maybe also reiterates that it's hard to make the interpretation stated without this knowledge at the earlier point in the manuscript}. Alternatively, the WRPRFa interaction with E79Q could be equal to WT and not diminished at all, and it simply be that the peptide has a different pharmacological outcome at E79Q than WT. I think the interpretation here either has to be caveated with alternative reasons, or simply left out for now and the results presented without interpretation (and this left to later). The interpretation here is also somewhat inconsistent with the stronger wording later in the paragraph stating that "our experiments demonstrate the lower glutamate pair (E78/E421) is important for WRPRFa activity."

We rearranged Figure 4 and the accompanying Results section to present the effect of the E78Q mutation on pH-sensitivity earlier. We also restated our conclusions on the relative roles of the lower glutamate pair in interaction with WRPRFa. [317-353]

Minor comments:

- Lines 43-44: ASIC2b homomers are also not thought to open in response to extracellular acidification, and may be worth noting like has been done for ASIC5 here.

We updated the sentence to remove mention of ASIC5/BASIC and added references to support the acid insensitivity of ASIC2b and ASIC4. [41-43]

“The ASIC family is composed of four genes, ASIC1-4, that give rise to six distinct isoforms: ASIC1a, 1b, 2a, 2b, 3, and 4. All isoforms respond to extracellular acidification except homomeric ASIC2b (3) and ASIC4 (4).”

- Line 50: The “wrist” is more typically thought of as the region linking the transmembrane domain and ECD, not so much that the transmembrane domains are the wrist as described in text.

We emended this sentence to correctly identify the wrist. [47-49]

“Between the two transmembrane segments is a large extracellular domain (ECD) whose structure has been described as a “closed-fist” with the transmembrane domains connected to the ECD by a “wrist”.”

- Line 72-78: When introducing the ASIC3 targeting molecules, both APETx2 and RPRFa are described. Adding the earlier FMRFa paper (PMID 10798398) and the more recent conoRFamides (PMID 31028742) seems to fit here. Particularly as both papers describe selectivity within ASIC subtypes.

We included both recommended references in the text as additional examples of RFamides exhibiting selectivity between ASIC isoforms. [77-79]

“Other RFamides exhibit lower margins of selectivity, like FRRFa and FMRFa (32), or can exert opposite effects on different isoforms, like As2a (33).”

- Fig S1: The figure legend has “I3s/lpk” but the graph y-axis is “I10s/lpk”. If the figure legend is incorrect, to correct the typo.

We corrected the figure legend. [1202-1203]

- Supp Fig legends: All the Supp Fig legends have “ \times/\div ” where it’s a typo or conversion error for “ \pm ”? This also appears in the main text at points (e.g. Line 213 and 217).

We used \times/\div to indicate the geometric standard deviation and \pm for the arithmetic standard deviation. We updated the statistics section of Methods to make this clearer. [824-828].

“In cases where averages are plotted, points represent geometric means and error bars geometric standard deviation (geometric mean \times/\div geometric standard deviation). All

averages stated in the text or tables are geometric means with geometric standard deviations except when the data set contained negative values and we instead report arithmetic mean with standard deviation (arithmetic mean \pm standard deviation).”

- Line 114: “higher than previously published (4.23 μ M) (29)” it might be good to add a qualifier to this sentence such as “higher than previously published (4.23 μ M) under similar conditions” – given the cited data is analysing I5s/Ipeak and not I3s.

We included a sentence to provide possible explanations for the observed difference in potency. [115-120]

“The discrepancy in EC₅₀ could arise from species or methodological differences: our study measured the fraction of sustained current at 3 seconds post-peak using human ASIC3 while the prior study used rat ASIC3 and measured sustained current at 5 seconds.”

- Line 1052 (Fig S2 legend): could be worth noting in the legend that the QPatch II was used for this comparison. Just to make clear it’s not local perfusion as would be possible with the manual rig setup.

We updated the legend to indicate testing was performed on the QPatch II. [1210]

- Fig 1G-I: Please confirm/clarify that this data for WRPRFa is using the same setup as in Fig 1A-C, and not the QPatch II.

We updated the legend for Figure 1 and the Results section to clearly indicate which experiments were performed on MPC vs QPatch II. Furthermore, we added sentences to the Results and Methods sections indicate all experiments were performed by MPC as a default, except when specifically indicated to have been done on QPatch II. [749-751, 768-769]

- Fig S6: The axis labels are missing.

We added the missing labels to the axes.

- Line 277-278: Regarding the lack of measurable currents from the glutamate to aspartate mutants, I think it would help to either note in text or as a figure the pH conditions used for these experiments. It could just be that the pH-activation threshold is shifted for the aspartate mutants.

We added a sentence addressing the possibility of shifted proton sensitivity as well as poor expression or trafficking. [301-304]

“This could be due to poor expression or problems of surface trafficking, or because the proton sensitivity of the mutant channels was shifted such that the pH 6.3 stimulus we used was insufficient to activate the channels.”

- Line 290-299: If space permits, it could be helpful to the reader to include the typical square WT/WT, WT/MT, MT/WT, MT/MT quadrant box for two double mutant cycle analyses pairs. Potentially easier to read/follow than the text alone and having to draw the entire mental map.

Multiple reviewer comments addressed elements of our interaction site mutagenesis study and the pH sensitivity change for E78Q. We split Figure 4 into 2 figures, which allowed us to address the effect of the E78Q mutation on pH-sensitivity in Figure 4 while Figure 5 presents the double mutant cycle analysis results. This also provided space to add quadrant boxes for both mutant pairs in Figure 5. Additionally, we added Table 2, containing parameters for concentration-response curves of different peptide/channel combinations, which increased the readability of the text.

- Line 326-328: Regarding the concatemer channels. Agree that most previous studies have used co-injection or co-transfection, however, similar concatemeric channel designs have been used before. PMID: 27277303, 37422581, or 35156612 for similar ligand binding/stoichiometry studies.

We included the recommended references as prior examples of ASIC concatemers used to control subunit stoichiometry. [370-372]

Reviewers' comments:

Reviewer #1 (Remarks to the Author):

I appreciate that the authors have substantially improved their manuscript. I have the following remaining comments:

We thank Reviewer #1 for their feedback.

1) The last part of the title (“..., which unveils gating mechanisms.”) is no longer appropriate, and the authors should consider changing it.

We removed the last part of the title. [5]

2) Lines 397-399: “...these results suggest a single ASIC3 subunit contains a sufficient WRPRFa binding site...”. I do not agree with this conclusion because it suggests three independent binding sites. It would be safer to say “...these results suggest all three ASIC3 subunits contribute to a WRPRFa binding site...”. The same for lines 565-566

We cannot say with certainty whether there are 3 independent binding sites (and only one occupied) or a single site formed by 3 subunits. To prevent misunderstandings, we changed “contains” to “provides” [355, 401, 574]. This removes any implication of discrete binding sites while maintaining the conclusion that a single ASIC3 subunit confers sensitivity to WRPRFa – whether by providing 1 of 3 binding sites or providing 1/3rd of a single binding site, either being sufficient for partial activity. Additionally, we added the following sentence to the Discussion:

“However, maximal activity of WRPRFa likely requires a full binding site formed by 3 ASIC3 subunits.” [577-578]

3) Figures 7C-E: I do not agree with the interpretation of these results. In Fig. 7C, peak ii is clearly different from peak i, indicating that WRPRFa unbound in between. The ratio I_{3s}/I_{pk} is an insufficient parameter, as both I_{pk} and I_{3s} decreased between peaks i and ii. If one would fit the decay of peak i, the decay of peak ii would nicely follow this fit, suggesting that WRPRFa unbound during decay.

I also note that the very slow washout of WRPRFa from the open state suggests that the peptide is trapped in its binding site. Otherwise, if binding/unbinding were simply limited by diffusion, such a slow wash-out would suggest a much higher affinity. Binding site trapping would be compatible with binding to a cavity in the lower palm domain.

We agree that both I_{pk} and I_{3s} are lower in peak ii than peak i, but their proportions are the same, indicating the fraction of sustained current is the same in peak i and peak ii. This is in stark contrast to the outcome with RPRFa, where the fraction of sustained current is significantly lower in peak ii than peak i, demonstrating that some of the channels became

unbound during the peak i decay process. The I_{3s}/I_{pk} metric we use is very similar to the $I_{2.5s}/I_{pk}$ metric used by Reiners et al.; we believe using the same experimental protocol and same analysis method makes the distinct effects of RPRFa and WRPRFa even more compelling.

We agree that a fit to the decay of peak i would likely follow the decay of peak ii, but we do not see that as evidence WRPRFa unbinds during decay. Instead, this may better describe the slow desensitization step from O_B to D_B for WRPRFa vs the unbinding of RPRFa from O_B to O .

4) Lines 552-553: the sentence starting with “Alternatively, ...” is confusing. Please rephrase.

We rephrased this sentence as follows:

“Alternatively, the RFamide binding site in the activated channel could be “shut”; this would prevent WRPRFa from reaching its binding site as well as slowing its dissociation from the channel” [563-563]

Minor: Line 77: change “As2a” to “ASIC2a”

As2a is an RFamide that shows distinct effects on ASIC1 and ASIC3.

Reviewer #2 (Remarks to the Author):

The authors have sufficiently addressed most of my concerns. There are a few remaining points, that I list here by using the previous numbering of the comments.

We thank Reviewer #2 for their attention to detail and help refining the manuscript.

Major points

2.5. Please provide more specific y-axis titles also for Figs. 4F, 4G, 8A and 8C.

We updated the y-axis titles.

3.2. There is a discussion of possibly changed pKa values of residues E78 and E421 (lines 574-586). A change in pKa should also affect the pH dependence. For this reason it is important to discuss the evidence for shifts in pH dependence of mutations at these positions (PMID 17389250 and PMID 22948146).

We added a sentence discussing the effect of mutations on channel pH-sensitivity and made reference to the provided citations.

“Disturbance of the pKa shift would alter the pH-dependence of side chain charge, consistent with changes in channel pH-sensitivity previously reported for mutations of the lower glutamate pair in ASIC1a and ASIC3 (61, 62).” [591-594]

5. Lines 354-355, "Given the E78/E421 pair is located away from subunit interfaces...", seems wrong to me, since these residues are oriented towards the central cavity, where the three subunits approach each other.

We agree the glutamate pairs from each subunit do point towards the central cavity, but our statement is meant to indicate the glutamate pair is not part of a subunit-subunit junction. We have rephrased the sentence to clarify this ambiguity:

“Given the E78/E421 pair is not located at a subunit-subunit interface, it may be possible that most of or the entire WRPRFa binding site could be provided by a single subunit” [354-355]

6. Lines 614-621, when discussing previous studies measuring tachyphylaxis, mention also that in the present study, a higher stimulation frequency was used than in previous studies.

We added this statement:

“Here, we also used a higher stimulation frequency than prior studies.” [631]

New analysis in Fig. 7C-D, lines 419-445. The description of the experiment is difficult to follow; I had to look it up in Reiners et al. to completely understand. Could you reformulate it in a way that it is understandable without going back to the original description of the protocol? In the text lines 443-445, when making conclusions about changes in the peak amplitude in this protocol, provide data. In the legend to Fig. 7C, indicate the timing, since this is not so clear from the figure itself.

We rewrote this section to make the experiment clearer, included below. Additionally, we added our data repeating the Reiners et al. experiment as Figure S9B and C to facilitate comparison of the distinct peptide effects without need to visit another reference. We updated Figure S9D to include the I_{pk} and I_{3s} values for peak ii and peak iii normalized to peak i. We updated the Figure 7C legend to indicate timing.

“We performed an experiment like that done by Reiners et al. to ask if WRPRFa would function the same way as RPRFa; that is, the peptide must first dissociate from the channel before desensitization can occur. HEK cells expressing ASIC3 were pre-treated with 30 μ M WRPRFa to load channels with peptide, then activated by a pH 5.0 solution without peptide, which induced a relatively rapid desensitization. If WRPRFa must first unbind from channels before desensitization, then continuous flow of pH 5.0 solution would wash away peptide that dissociated from channels prior to desensitization. Once most of the current was gone, we recovered the channels for 2 seconds at pH 8.0, which

allowed unbound and desensitized channels to partially recover (Fig. S9A). After the brief recovery interval, we resumed the pH 5.0 stimulus to re-activate unbound channels that recovered during the 2 second interval, as well as channels still bound to peptide.

If WRPRFa dissociated from channels before desensitization, then the peptide would have been washed away during the peak i stimulus period. Thus, fewer channels would be peptide-bound in peak ii and the fraction of sustained current would be lower for peak ii than for peak i. Unexpectedly, we saw no difference in the I_{3s}/I_{pk} between peaks i and ii, indicating that desensitization of WRPRFa-bound channels did not result in an increased number of unbound channels (Fig. 7D). This is a different outcome than was seen for RPRFa (Fig. S9B and C) (45). For RPRFa, both in the previous study and our study, peak ii showed a decreased sustained current fraction compared to peak i, reflecting an increased proportion of the fast-desensitizing component originating from unbound channels.” [425-445]

Small changes to make (new numbering)

1. Line 77, would "As2a" mean "ASIC2a?"

As2a is an RFamide that shows distinct effects on ASIC1 and ASIC3.

2. Lines 390-391, "The shift in pH-sensitivity for ASIC313 was less than WT ASIC3 (0.73 units), and ASIC131 was shifted even less than ASIC313, consistent with the number of ASIC3 subunits in the channel." Provide evidence that these values are statistically different from each other.

We included the results of the AICc comparison to indicate the probability the two parameters are different, as done elsewhere in the manuscript.

“The shift in pH-sensitivity for ASIC313 was less than WT ASIC3 (0.38 units vs 0.73 units; ASIC313 vs WT ASIC3, >99.9% pH_{50} shift is different), as was the shift for ASIC131 (0.26 units vs 0.73 units; ASIC131 vs WT ASIC3, >99.9% pH_{50} shift is different). ASIC131 was shifted even less than ASIC313 (0.26 units vs 0.38 units; ASIC131 vs ASIC313, >99.9% pH_{50} shift is different), consistent with the number of ASIC3 subunits in the channel.” [390-395]

3. Lines 393-397, similarly to the passage above, provide results of statistical tests to support these statements.

We included the results of the AICc comparison to indicate the probability the two parameters are different, as done elsewhere in the manuscript.

“Furthermore, the EC_{50} for WRPRFa also changed with the number of ASIC3 subunits in the channel. WRPRFa was less potent on ASIC313 (lower limit EC_{50} : 12.6 μ M, 95% CI: 10.2 to 15.6 μ M; vs ASIC3, >99.9% EC_{50} are different) than ASIC3 and even less so

on ASIC131 (lower limit EC₅₀: 32.4 μM, 95% CI: 28.5 to 36.5 μM; vs ASIC3, >99.9% EC₅₀ are different)” [398-399]

4. Mutant cycle analysis, lines 321-351, WIPRFa is used for testing position R4, while WRP{ADMA}Fa is used for R2. For the analysis of the position E378/E416, WIPRFa is used. Therefore, on Line 347, "the site 2 Arg with.." should rather be "the site 4 Arg with..".

Thank you for finding this error. We updated “site 2” to “site 4” [347, 349, 350]

Reviewer #3 (Remarks to the Author):

Thank you to the authors for considering all the reviewer comments. The updated manuscript looks good. Hopefully future work will resolve the still contentious points that have been uncovered by the data in this manuscript.

We thank Reviewer #3 for their contribution and hope they share our curiosity for what future RFamide/ASIC studies will discover.

REVIEWERS' COMMENTS:

Reviewer #2 (Remarks to the Author):

The authors have now sufficiently addressed the points that have remained unresolved after the first revision.

We thank Reviewer #2 for their review of our work.